# Preventing acute asthmatic symptoms by targeting a neuronal mechanism involving carotid body lysophosphatidic acid receptors

Nicholas G. Jendzjowsky [1], Arijit Roy[1], Nicole O. Barioni[1], Margaret M. Kelly[2,3], Francis H. Y. Green[2,3], Christopher N. Wyatt[4], Richard L. Pye[4], Luana Tenorio-Lopes[1] & Richard J. A. Wilson[1]

Asthma accounts for 380,000 deaths a year. Carotid body denervation has been shown to have a profound effect on airway hyper-responsiveness in animal models but a mechanistic explanation is lacking. Here we demonstrate, using a rat model of asthma (OVA-sensitized), that carotid body activation during airborne allergic provocation is caused by systemic release of lysophosphatidic acid (LPA). Carotid body activation by LPA involves TRPV1 and LPA-specific receptors, and induces parasympathetic (vagal) activity. We demonstrate that this activation is sufficient to cause acute bronchoconstriction. Moreover, we show that pro-phylactic administration of TRPV1 (AMG9810) and LPA (BrP-LPA) receptor antagonists prevents bradykinin-induced asthmatic bronchoconstriction and, if administered following allergen exposure, reduces the associated respiratory distress. Our discovery provides mechanistic insight into the critical roles of carotid body LPA receptors in allergen-induced respiratory distress and suggests alternate treatment options for asthma.

[1] Department of Physiology and Pharmacology, Hotchkiss Brain Institute, Alberta Children's Hospital Research Institute, Cumming School of Medicine, University of Calgary, 3330 Hospital Drive NW, Calgary Alberta T2N 4N1, Canada. [2] Department of Physiology and Pharmacology, Snyder Institute for Chronic Diseases, Cumming School of Medicine, University of Calgary, 3330 Hospital Drive NW, Calgary Alberta T2N 4N1, Canada. [3] Department of Pathology and Laboratory Medicine, Snyder Institute for Chronic Diseases, Cumming School of Medicine, University of Calgary, 3330 Hospital Drive NW, Calgary Alberta T2N 4N1, Canada. [4] Department of Neuroscience, Cell Biology and Physiology, Wright State University, 3640 Colonel Glenn Hwy, Dayton, OH 45435-0001, USA. Correspondence and requests for materials should be addressed to R.J.A.W. (email: wilsonr@ucalgary.ca)

Asthma is one of the most common lung diseases, affecting 241 million people worldwide and is the cause of 380,000 deaths per year[1]. Asthma is characterized by airflow limitation caused by inflammation, excess mucous secretion, remodeling changes, such as goblet cell metaplasia and increased smooth muscle mass and acute conducting airway constriction in response to stimuli, usually allergens (i.e., bronchoconstriction)[2]. Acute bronchoconstriction is usually treated with inhaled short-acting β2-agonists, corticosteroids, and/or ipratropium bromide, but inhaled drugs have only limited value in situations of severe bronchoconstriction and mucus plugging, when access to their site of action is blocked. Consequently, acute bronchoconstriction remains the leading cause of hospitalization and asthmatic sudden death, dictating the need for new medications[2]. Our understanding of how allergens trigger asthma is advancing rapidly[2], but how immune responses cause bronchoconstriction remains an important knowledge gap.

All lungs demonstrate acute airway bronchoconstriction in response to irritants such as capsaicin, but only asthmatic lungs respond acutely to allergens and/or bradykinin[3–10]. The effects of allergens/bradykinin on asthmatic lungs is not simply the release of local inflammatory mediators activating strictly local reflexes[4–13] but likely involves a circuit that includes the brainstem because vagotomy annuls allergen-induced bronchoconstriction in animal models[12,14]. This circuit may be the lung-vagal afferent to parasympathetic efferent reflex pathway that mediates the effects of irritants in naive lungs[11,12,14] or, as carotid body activation elicits bronchoconstriction, the afferent arm of this reflex may originate at the carotid body[15–23]. Notwithstanding this possibility, early (and now largely abandoned) attempts to use unilateral carotid body resection in over 5000 humans as a treatment for asthma are not supported by clinical trials[24] and to date, there is no conclusive evidence for the therapeutic effects of the more risky, but possibly more efficacious bilateral resection in humans[25,26]. Further hindering the acceptance for a role of the carotid body in asthma, no feasible mechanism has emerged linking enhanced activity to the asthmatic lung[27–29].

A growing realization of the importance of the carotid bodies in sleep apnea and cardiovascular diseases has led to a resurgence of interest in their properties[30–32]. We recently discovered that the exquisite heat sensitivity of the carotid body is mediated in large part by transient receptor potential cation channel vanilloid 1 receptors (TRPV1) in axons of chemosensory afferents (cell bodies in the petrosal ganglia, post-synaptic to oxygen-sensing glomus cells)[33]. However, the physiological significance of this observation was unknown. TRPV1 is a multimodal sensor capable of responding to heat, pH, anandamide, and inflammatory mediators including IL-1[34] and lysophosphatidic acid (LPA)[35,36]. As LPA has garnered attention as an important biomarker of the immune response in the murine house dust mite model of asthma and is present in human bronchiolar lavage fluid and venous samples following asthmatic challenge[37], we hypothesize that LPA released from the asthmatic lung in response to allergen/bradykinin challenge activates TRPV1 in the carotid body causing acute bronchoconstriction.

Here we report that LPA, at concentrations present in the systemic circulation following allergen or bradykinin challenge, triggers acute allergen/bradykinin-induced bronchoconstriction via activating carotid body TRPV1 and LPA-specific receptors. Moreover, we demonstrate that blocking this pathway with systemic TRPV1 and LPA-specific receptor antagonists ameliorates this effect. Thus, our data provide a missing link between carotid body-mediated bronchoconstriction and allergen-induced asthma. We suggest TRPV1 and LPA receptor blockade may provide a systemic alternative to conventional inhaler-based therapy for acute allergen-induced bronchoconstriction that does not require drug access through compromised airways.

## Results

**LPA activates TRPV1 and LPA receptors in the carotid body**. LPA activates TRPV1 receptors[35,36]. Previously, we demonstrated that TRPV1 receptors are expressed in the terminals of petrosal neurons that innervate the carotid body (i.e., the axon terminals of chemosensory afferent[33]) but LPA also binds to 6 LPA-specific G-protein coupled receptors (GPCR), LPAr 1 through 6. To test for the presence of these receptors in the carotid body, we used RT-PCR. cDNA for LPAr 1, 3, 4, and 6, were present in the carotid body; LPAr 3 and 6 were also in petrosal ganglia; and LPAr 1, 3, 4, and 6 were present in the superior cervical ganglia. We found no evidence for expression of LPAr 2 in these carotid body associated tissues (Fig. 1a, Supplementary Fig. 1a).

Next, we tested whether LPA (18:1 unless otherwise stated) has functional effects on glomus cells using Fura 2 calcium imaging. LPA (5 μM) increased intracellular calcium release producing calcium spikes of similar order of magnitude to that produced by 20 mM potassium (Fig. 1b). Building on this, we tested whether common LPA species present in blood (16:0, 18:1, and 18:2) have functional effects on the carotid body output using an en bloc perfused carotid body preparation (Fig. 1c)[33,38]. All LPA species tested had a dose dependent effect on carotid sinus nerve activity (Fig. 1d, e). To determine which receptors mediate this, we tested the effects of LPAr blockade (BrP-LPA, 1.5 μM or Ki16425, 5 μM), TRPV1 blockade (AMG9810, 10 μM), and dual blockade (Fig. 1f–h, Supplementary Fig. 1b-e). AMG9810 reduced 5 μM LPA-mediated carotid sinus nerve excitation by 41 ± 6%; BrP-LPA or Ki16425 reduced 5 μM LPA-mediated excitation by 70 ± 3% or 61 ± 9%, respectively; and dual blockade with AMG9810 and either BrP-LPA or Ki16425 reduced excitation by 77 ± 5% or 89 ± 3%, respectively (Fig. 1f–h, Supplementary Fig. 1b-e). These data demonstrate LPA stimulation of the carotid body involves both TRPV1- and LPA-specific GPCR's.

**Carotid body activation by LPA causes vagal efferent activity**. To test if LPA activation of the carotid body stimulates chemoafferents (not baro-afferents) and, is capable of inducing parasympathetic (vagal) activity with the capacity to cause bronchoconstriction, we turned to the in situ dual-perfused preparation (Fig. 2a). This preparation allows artificial perfusion of the carotid bodies and brainstem independently, while recording from phrenic and vagal efferents[39]. Once baseline conditions were established (brainstem perfused with 40 Torr $PCO_2$, balanced with $O_2$; carotid bodies perfused with 35 Torr $PCO_2$ and 100 Torr $PO_2$, balanced with $N_2$; Supplementary Fig. 2a), we tested carotid body viability using a 10 min bout of hypoxic perfusate (50 Torr $PO_2$ and 40 Torr $PCO_2$ balanced with $N_2$, Supplementary Fig. 2a). Following recovery, the brainstem perfusate was switched to hypocapnia ($PCO_2 ≤ 20$ Torr, balance $O_2$). Central hypocapnia induces a state of complete apnea (cessation of phrenic bursts > 1 min) and optimizes the sensitivity of the preparation to carotid body stimuli[40]. Next, a bolus of 5 μM LPA was delivered to the carotid body circulation. With carotid bodies intact, hypoxia and LPA induced increases in phrenic (indicating LPA recruits carotid body chemoafferents) and vagal nerve efferent discharge (Fig. 2b, d–f; Supplementary Fig. 2a, b; Supplementary Fig. 3a-d). With the carotid sinus nerve resected, the response to hypoxia (Supplementary Fig. 2a) and LPA (Fig. 2c, e, f; Supplementary Fig. 2c and Supplementary Fig. 3a-d) was abolished.

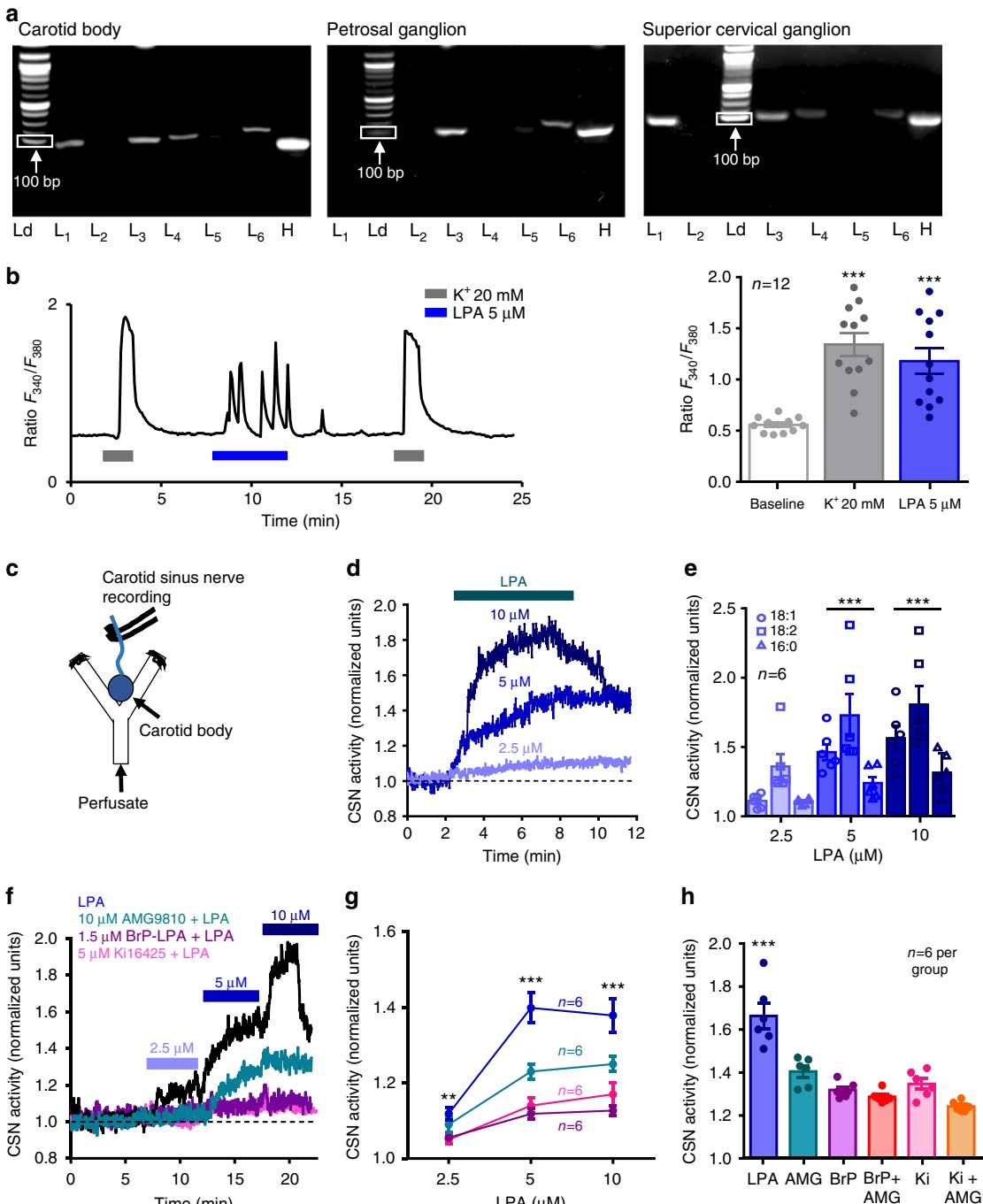

**Fig. 1** LPA stimulates the carotid body. **a** RT-PCR: LPAr (L) 1, 3, 4, and 6 in carotid body (left), LPAr 3 and 6 in petrosal ganglia (center), and LPAr 1, 3, 4, and 6 in superior cervical ganglia (right) in reference to a housekeeping gene, hypoxyribosyltransferase (H; cDNA ladder, Ld, was used to identify tissues used for each gel, 100 bp marker indicated). **b** Calcium imaging with Fura-2 loaded carotid body glomus cells (12 cells on 5 cover slips) reveals an increase of intracellular calcium in response to 5 μM LPA (18:1 unless otherwise stated) and 20 mM K+. $F_{2,35}$ (one-way ANOVA) = 18.220, $p < 0.001$; Holm–Šidák post hoc: difference from baseline, ***$p < 0.01$. **c** En bloc perfused carotid body preparation used to record chemosensory afferents in the carotid sinus nerve (CSN). **d** LPA 18:1 causes an increase in CSN activity. **e** The dose-dependent relationship differed slightly between several species of LPA found in blood (16:0, 18:1, and 18:2); $F_{4,53}$ (two-way ANOVA: species × dose) = 3.380, $p = 0.021$. Holm–Šidák post hoc: all species 5 and 10 μM different from 2.5 μM, ***$p < 0.001$. **f** Effects of TRPV1 and LPAr antagonists on CSN response to LPA. No blockade (black trace), TRPV1 blockade (AMG9810 10 μM, teal trace, or LPAr blockade (BrP-LPA 1.5 μM, mauve trace; Ki16425 5 μM, fuscia trace). **g** Summary data. Responses 2.5 μM LPA: $F_{3,23}$ (one-way ANOVA) = 5.031, **$p = 0.01$. Holm–Šidák post hoc: LPA different from BrP-LPA ($p = 0.026$) and Ki16425 ($p = 0.018$). Responses 5 μM: $F_{3,23}$ (one-way ANOVA) = 24.547, ***$p < 0.001$. Holm–Šidák post hoc: LPA different from BrP-LPA, Ki16425, and AMG9810 ($p < 0.001$); AMG9810 different from Brp-LPA ($p = 0.018$) and Ki16425 ($p = 0.044$); Responses 10 μM: $F_{3,23}$ = 14.231, ***$p < 0.001$. Holm–Šidák post hoc: LPA different from BrP-LPA ($p < 0.001$); Ki16425 ($p < 0.001$); AMG9810 ($p = 0.022$); AMG9810 different from BrP-LPA ($p = 0.022$). **h** Summary data of 5 μM LPA (blue), with TRPV1 blockade (AMG9810, 10 μM; teal), LPAr blockade (BrP-LPA, 1.5 μM; mauve; Ki16425, 5 μM; fuscia), or LPAr + TRPV1 blockade (AMG9810 + BrPLPA red; AMG9810 + Ki16425 orange), $F_{5,35}$ (one-way ANOVA) = 26.164, $p < 0.001$. Holm–Šidák post hoc: AMG9810 + LPA significantly different from Ki16425 + AMG9810 + LPA, $p = 0.005$; LPA significantly different from all other groups, ***$p < 0.001$. All data are presented as mean ± sem

**Carotid body activation by LPA causes bronchoconstriction**. To test if LPA activation of the carotid body is capable of causing acute bronchoconstriction in vivo, we performed lung function measurement in naive animals using an anesthetized artificially ventilated preparation with the Flexivent respirator system (Fig. 3a). Saline, 5 μM LPA, and 6 μM NaCN (an independent test of carotid body sensory viability)[41], were delivered consecutively to each animal (0.5 ml bolus injected over 1 min via vena cava catheter; 10 min between challenges). LPA and NaCN, but not saline, induced increased airway resistance in preparations with intact CSN (Fig. 3b, c). These challenges had no effect in preparations with bilateral carotid sinus nerve resection, demonstrating the necessity of the carotid body in LPA-induced bronchoconstriction (Fig. 3b, c).

**Arterial LPA is associated with bronchoconstriction**. If asthma-associated acute bronchoconstriction is caused by activation of the carotid bodies by LPA released from an allergen-challenged asthmatic lung, (a) the arterial concentration of LPA should increase significantly during acute inflammation and (b) the plasma from asthmatic rats should stimulate an isolated carotid body in a TRPV1 and LPA receptor dependent fashion. To test this, we used the ovalbumin-sensitized (OVA) Brown Norway rat model of asthma (Fig. 4a)[42]. We confirmed that this model exhibits many of the salient features of human asthma. These include an increased Inflammatory Index Score (i.e., elevated eosinophil cell counts in airways and bronchoalveolar lavage fluid (BALF, Supplementary Fig. 4e, f), increased airway smooth muscle thickening and epithelial goblet cell metaplasia (Fig. 4b–e, Supplementary Fig. 4a-d), increased baseline airway resistance (Fig. 4g) and increased gene expression of chemoattractant in lung tissue (Fig. 4f). While methacholine is often used to trigger robust acute bronchoconstriction in this model, we opted to use bradykinin because parasympathetic efferents are cholinergic, and thus exogenous methacholine is likely to mask parasympathetic involvement. Bradykinin delivered on day 28 (0.4 mg; nebulized) caused a marked increase in airway resistance in OVA but not naive rats within 20 min (Fig. 4g). Arterial plasma samples were analyzed with ELISA prior to and ~20 min following bradykinin stimulation (Fig. 4h, Supplementary Fig. 5). Bradykinin had no effect on plasma concentration of LPA in naive rats but significantly increased LPA in the OVA group (Fig. 4h, Supplementary Fig. 5). The coefficient of variation for all duplicate samples was 2.4 ± 0.3%. In a subset of samples, μM concentration of total LPA in plasma from OVA rats following bradykinin was confirmed by liquid chromatography–mass spectrometry (LC/MS; Supplementary Fig. 6).

To test if the LPA in plasma from asthmatic rats is sufficient to stimulate the carotid body, we harvested plasma from naive and asthmatic animals 3 h after allergen challenge and tested their effects on carotid sinus nerve activity in the isolated en bloc carotid body preparation. Naive and asthma plasma caused 16 ± 2% and 39 ± 3% increases in carotid sinus nerve activity, respectively. Dual blockade of TRPV1 and LPA receptors with AMG9810 and Brp-LPA reduced the response to asthma plasma by 79 ± 2% (Fig. 5a, b).

**LPA-carotid body axis induces asthmatic bronchoconstriction**. In order to examine the involvement of LPA-induced carotid body activity on airway resistance in response to bradykinin nebulization we used alfaxan-anesthetized animals and the Flexivent system. These experiments were performed with a number of acute interventions affecting the proposed pathway, including vagotomy, carotid body denervation, and TRPV1 and/or LPA receptor blockade. Prior to bradykinin, these interventions had no effect on airway resistance ($p > 0.06$; Holm–Šidák post hoc test comparing asthma control vs any manipulation group) and as expected, bradykinin had no effect on non-asthmatic lungs ($p > 0.3$; Holm–Šidák post hoc test comparing asthma control vs any manipulation group). However, the acute bronchoconstriction induced by bradykinin in asthmatic lungs was diminished by at least 60% with all interventions (Fig. 6b, Supplementary Fig. 7). The loss of bradykinin-induced bronchoconstriction also occurred in chronic (5d) carotid body denervated, but not sham rats (Fig. 6c).

To ensure maintenance of vagal–vagal reflexes after carotid body denervation and thereby rule out the possibility of indirect effects of carotid body denervation, we tested whether carotid body denervation abolished bronchoconstriction induced by aerosolized capsaicin which is a potent activator of C-fiber-mediated vagal–vagal reflexes. As expected, capsaicin-induced bronchoconstriction was abolished by vagotomy but not affected by carotid body denervation (Fig. 6d, Supplementary Fig. 8a, b). Similar results were obtained in naive and asthmatic animals (Fig. 6d, Supplementary Fig. 8a, b). In addition, to ensure that increased plasma LPA was not provoking increased airway resistance via a vagal–vagal reflex, we exposed asthmatic rats to nebulized LPA. LPA had no immediate effect on airway resistance. An increase in airway resistance occurred after 30 min, but this was abolished by carotid body denervation (Supplementary Fig. 8c). Together, these data suggest the increase in lung resistance with nebulized LPA is dependent on the carotid body.

**Blocking LPA receptors reduces acute bronchoconstriction**. In order to determine if blocking LPA signaling can be used to limit the severity of acute allergen-induced respiratory distress in conscious animals, 12 OVA rats were exposed to an aerosol containing ovalbumin (150 mg, nebulized) to trigger acute asthmatic bronchoconstriction and 20 min later, injected with a cocktail of TRPV1 and LPAr (i.p. 10 μM kg$^{-1}$ AMG9810 and 3 mg kg$^{-1}$ BrP-LPA, in 0.5 ml, Fig. 7d) antagonists. After the injection, animals were placed in a whole-body plethysmograph to monitor breathing. Following ovalbumin, expiratory time (Te) increased and inspiratory time: expiratory time (Ti:Te) ratio decreased, indicative of expiratory difficulty associated with acute asthmatic bronchoconstriction (Fig. 7a–f)[43]. However, the increase in Te and decrease in Ti:Te after 100 min was reduced in dual-block treated animals (Ti:Te: −28.3 ± 3.4%; Te: −72.9 ± 2.3%) compared to vehicle-injected or sham groups (Ti:Te: −49.2 ± 4.8%; Te: −59.1 ± 3.1%, and Ti:Te: −55.0 ± 4.9%; Te: −62.7 ± 2.9%; Ti:Te: $F_{2,35} = 10.064$, $p < 0.001$; Te: $F_{2,35} = 6.495$, $p = 0.004$, one-way ANOVA, Fig. 7e, f), suggesting this treatment can be used to reduce the severity of acute allergen-induced respiratory disturbances. Furthermore, treatment given 3 days prior to a final ovalbumin challenge was also mitigating, suggesting a long-term beneficial effect of this treatment (Fig. 7g, h).

## Discussion

Acute bronchoconstriction is a hallmark symptom of asthma and is a key reason for asthmatic hospitalization. Here we demonstrate that acute allergen/bradykinin-evoked bronchoconstriction and respiratory disturbances, in an animal model of asthma, involve the carotid body. Increased airway resistance from carotid body induced vagal stimulation is directly linked to activation of LPAr and TRPV1 receptors within the carotid body and, post synaptically, on petrosal ganglia axons. Our data suggest blocking LPA signaling ameliorates bradykinin-invoked bronchoconstriction in anesthetized asthmatic animals. Further, in conscious animals, where the severity of respiratory disturbances are fully

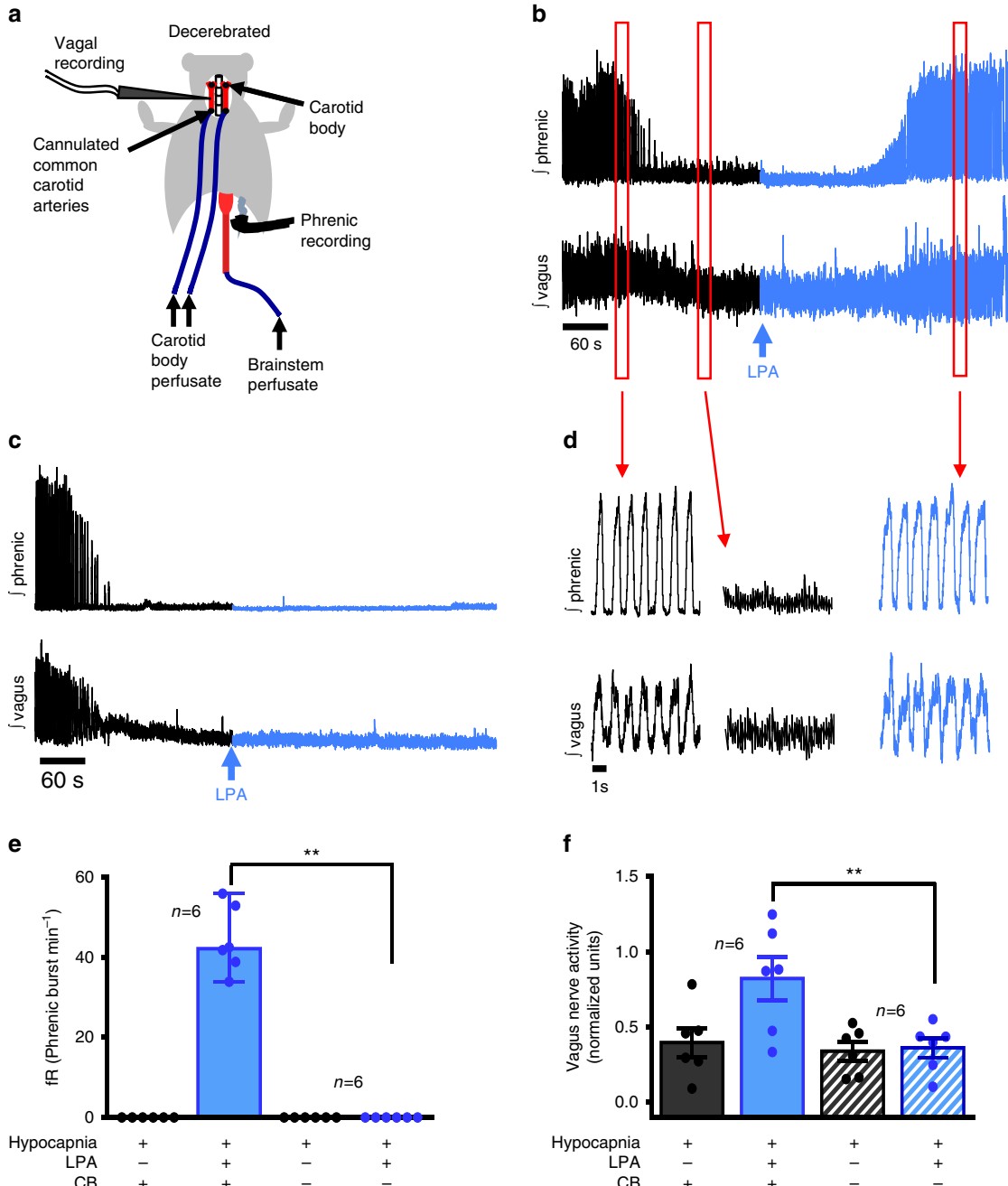

**Fig. 2** LPA stimulation of the carotid body increases efferent vagal activity. **a** The dual perfused in situ preparation used to record phrenic and vagal activity in response to specific carotid body stimulation. **b** Phrenic (upper trace) and vagal (lower trace) activity with the carotid body intact under brainstem hypocapnia (black trace) causing cessation of phrenic firing (neural apnea) and vagal quiescence. Activity is restored following carotid body stimulation with 5 μM LPA (blue arrow/trace), with restoration of vagal activity. **c** In denervated preparations the stimulatory effect of LPA on vagal and phrenic activity is absent. **d** Expanded sections of traces from **b** indicated by the red boxes, illustrating phrenic and vagal activity prior to brainstem hypocapnia induced apnea, apnea and following LPA injections. **e**, **f** Summary data during hypocapnia in intact (solid) and carotid body (CB) denervated (crossed) preparations for phrenic frequency (Phrenic burst min$^{-1}$; Mann Whitney: $U = 0$, $p = 0.002$, ** median ± range) and vagal total activity (normalized to baseline normoxic activity, normalized units, independent $t$ test: $t_{10} = 4.008$, $p = 0.0025$, ** mean ± sem)

exhibited without the mitigating effects of anesthesia, the treatment remains effective. Thus, this study provides a mechanistic explanation for the importance of the carotid body in acute asthmatic symptoms and points to alternate therapeutic drug targets for emergency intervention.

The carotid bodies have been implicated in asthma[15–23] but a role for the carotid bodies is unsupported by clinical trials[24] and

bilateral carotid body resection in humans is not advised because it abolishes the hypoxic ventilatory response. Nonetheless, activating the carotid bodies causes bronchoconstriction in normal lungs[16–20] and in one study in an animal model of asthma[15], carotid body denervation is reported to reduce the severity of nebulized methacholine induced hyper-responsiveness. While recent focus on the carotid bodies has been related to their role in

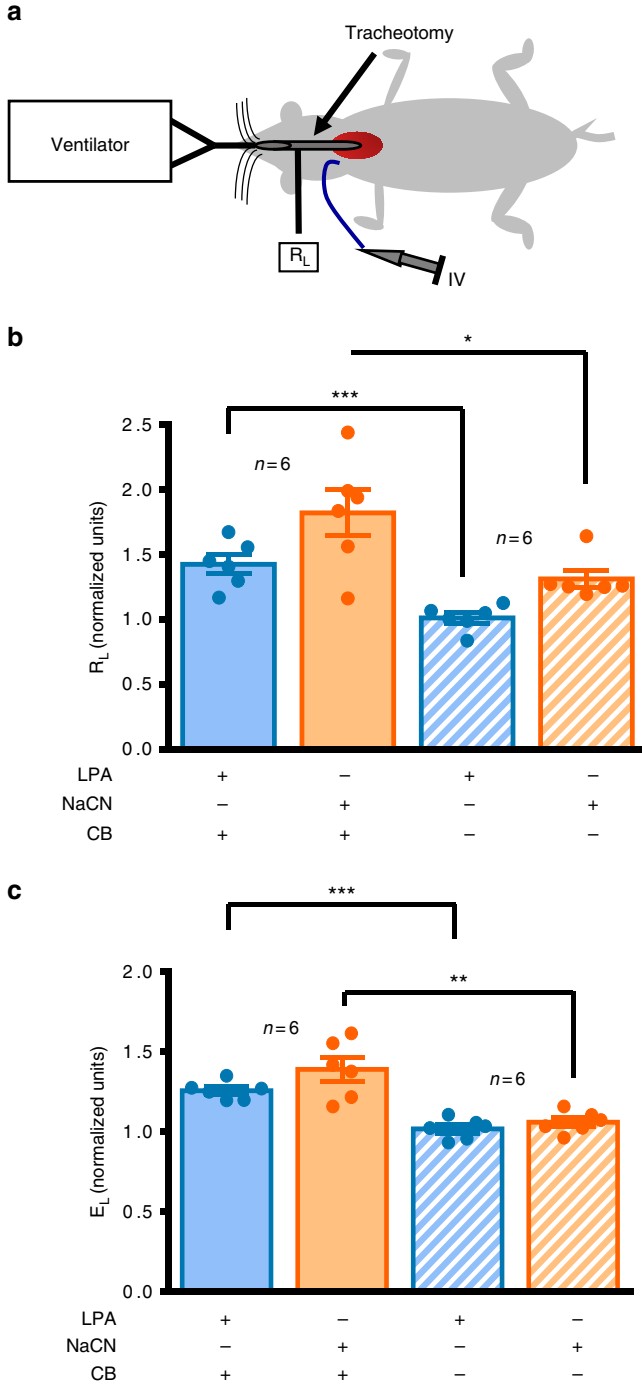

**Fig. 3** In vivo carotid body-mediated bronchoconstriction. **a–c** The anesthetized in vivo preparation used to measure airway mechanics in carotid sinus nerve intact (solid, $n = 6$) or denervated (crossed, $n = 6$) preparations, in response to jugular vein injection of lysophosphatidic acid (LPA 5 μM, blue) or sodium cyanide (NaCN, orange; an independent test of carotid body function). Baseline lung resistance ($R_L$) prior to LPA or NaCN injection: intact = $0.31 \pm 0.01$ cmH$_2$O s ml$^{-1}$; denervated = $0.28 \pm 0.02$ cmH$_2$O s ml$^{-1}$ (independent $t$-test: $t_{10} = 1.668$, $p = 0.126$). Baseline lung elastance ($E_L$) prior to LPA or NaCN injection: intact = $6.8 \pm 0.3$ cmH$_2$Oml$^{-1}$; denervated = $7.6 \pm 0.4$ cmH$_2$O ml$^{-1}$ (independent $t$-test: $t_{10} = 1.433$, $p = 0.182$). **b** $R_L$ in denervated compared to intact CB preparations—LPA-independent $t$-test: $t_{10} = 4.933$, \*\*\*$p < 0.001$; NaCN-independent $t$-test: $t_{10} = 2.711$, \*$p = 0.022$. **c** $E_L$, in denervated compared to intact CB preparations—LPA-independent $t$-test: $t_{10} = 6.772$, \*\*\*$p < 0.00001$; NaCN-independent $t$-test: $t_{10} = 4.202$, \*\*$p < 0.01$. In **b** and **c**, $R_L$ and $E_L$ are normalized to saline (baseline) measurements and presented as mean ± sem

ameliorated by carotid body denervation and/or vagotomy. We also show that bradykinin and allergen increase plasma LPA in asthmatic (but not naive) rats and that this plasma is capable of stimulating an isolated carotid body via an LPA receptor-dependent mechanism. These data prove the existence of a carotid body-mediated vagal reflex capable of increasing vagal efferent activity causing bronchoconstriction in response to LPA. As the carotid bodies are a main driver of ventilation, a role in mediating bronchoconstriction appears counterintuitive, however this mechanism may be responsible for maintaining rigidity of airways during increased ventilation and cough, and/or minimizing dead-space ventilation[44,45].

The substantive data above demonstrate a role for the carotid body-vagal reflex as a trigger for acute bronchoconstriction in an animal model of asthma. Arterial hypoxemia caused by poor lung function during an asthmatic attack was suspected as a trigger of the carotid body in asthma[46,47]. Indeed, acute bronchoconstriction caused by bradykinin resulted in hypoxemia in the OVA model (supplementary Table 1). However, in asthmatic humans, the severity of respiratory distress does not always correlate with arterial hypoxemia[48,49], and others have demonstrated no bronchoconstriction with reduced inspired O$_2$[27,50], suggesting additional and/or different carotid body triggering mechanisms.

Notwithstanding a possible role for hypoxia, our data demonstrates that the carotid body is activated by μM concentrations of LPA. LPA, produced from phosphatidic acid by phospholipases and from catalysis of lysophosphatidylcholine by autotaxin[51,52], is a central player in allergen induced asthmatic lung inflammation[53,54]. Several species of LPA are upregulated in the lung of asthmatic humans 48hrs after allergen challenge[37,55] and regulate prostaglandin levels, expression of Th2 cytokine receptors and IL13 signal transduction[56]. LPA may also increase the sensitivity of airway smooth muscle independently[57] and has been shown to augment acetylcholine mediated airway smooth muscle contractility[58].

While still an area of active research, LPA may also increase systemically. No studies have measured plasma LPA in humans immediately after an allergen challenge, but there is some evidence for a slight elevation 48 h after allergen challenge[37]. In our OVA model, we demonstrated that LPA in arterial plasma doubles (measured by ELISA[59–61]) 20 min after bradykinin but the dynamics of LPA are unknown[62,63]. Our estimates of LPA concentration 20 min following bradykinin (bradykinin caused an increase in lung resistance within 1 min) were between $1.5 \pm 0.2$ μM and $8.4 \pm 1.7$ μM as measured by LC/MS and ELISA, respectively (Supplementary Fig. 6). Given the half-life of LPA

exciting sympathetic activation, expected to cause bronchodilation via beta-receptor activation, we used the dual-perfused in situ preparation[39] to show that specific carotid body stimulation also causes an increase in vagal (presumably parasympathetic and acetylcholinergic) activity likely capable of causing bronchoconstriction. Nebulized methacholine, used as the provocation in most animal studies of asthma, is likely to short circuit this parasympathetic pathway, possibly explaining why its importance has been underestimated in asthma. In our study we induced acute asthmatic symptoms with allergen or bradykinin; both of which only cause pronounced bronchoconstriction in asthmatic lungs.

We show that bradykinin produces a ~threefold increase in lung resistance in asthmatic (but not naive) rats that was

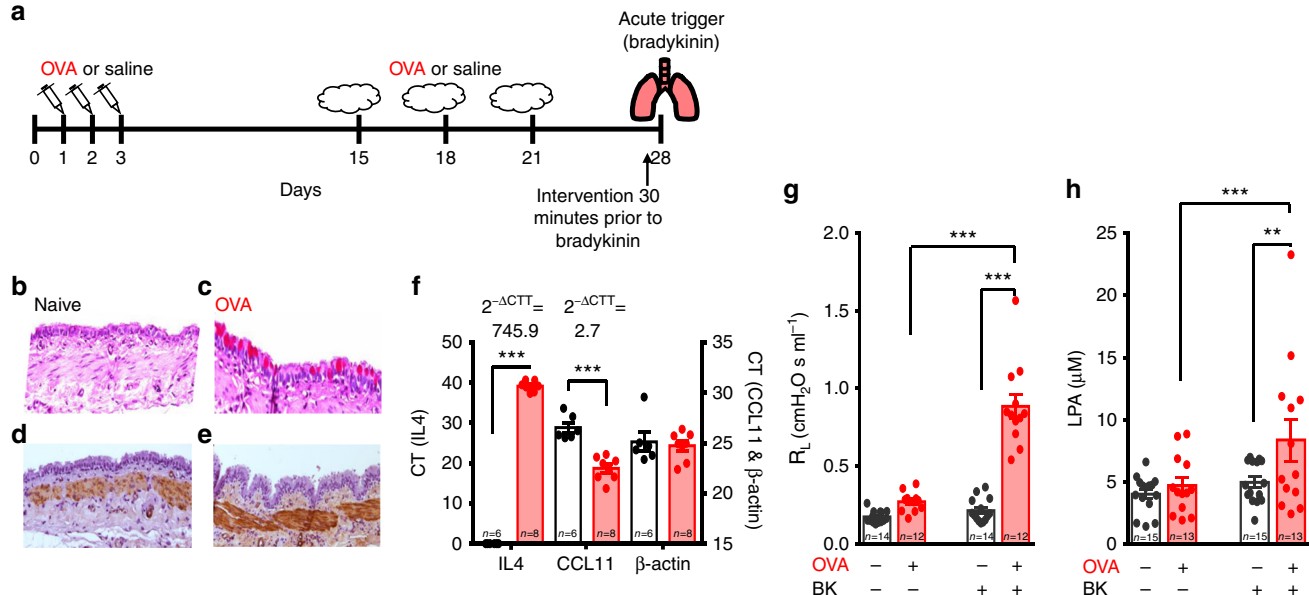

**Fig. 4** Ovalbumin-sensitized rats demonstrate increased airway remodeling, inflammatory markers, airway resistance and plasma LPA. **a** OVA-sensitization protocol (see Methods, OVA Cohort 1). **b–e** Typical asthmatic airway disease occurred in the OVA-sensitized rats (right **c**, **e**) compared to the naive group (left **b**, **d**) as indicated by the heightened presence of goblet cells (appearing pink with Periodic acid-Schiffs reagent; **b**, **c**) and thickening of airway smooth muscle (appearing brown when immuno-stained for smooth muscle actin; **d**, **e**). **f** qPCR of IL4, eotaxin (CCL11) and house-keeping gene β-actin for OVA (red) and naive (open) rats with calculated change in gene expression magnitude ($2^{-\Delta CTT}$[68]). CT: cycles threshold. OVA vs naive CT for IL4-independent $t$-test, $t_{11} = 73.271$, ***$p < 0.00001$; CCL11-independent $t$-test: $t_{12} = 6.369$, ***$p < 0.0001$; and β-actin-independent $t$-test: $t_{12} = 0.403$, $p = 0.694$. **g** Bradykinin increases $R_L$ in OVA animals only (red; $F_{1,51}$ (two-way RM ANOVA) = 69.224, $p < 0.001$). Holm–Šidák post hoc: bradykinin increases $R_L$ in OVA animals, ***$p < 0.001$; but not naive rats ($p = 0.40$). Prior to bradykinin $R_L$ in naive (open) and OVA rats is not significantly different ($p = 0.08$), following bradykinin $R_L$ is greater in OVA compared to naive rats (***$p < 0.001$). **h** Bradykinin increases LPA in OVA animals only ($F_{1,55}$ (two-way RM ANOVA) = 6.19, $p = 0.02$). Holm–Šidák post hoc: bradykinin increases LPA in OVA animals, ***$p < 0.001$; bradykinin has no effect on LPA in naive rats ($p = 0.21$), LPA in naive and OVA rats prior to bradykinin are not significantly different ($p = 0.58$); but LPA in OVA rats is greater than naive rats following bradykinin (**$p = 0.01$) (**g**, **h** See Methods, OVA Cohort 1). All data are presented as mean ± sem

(≤5 min)[62] and the rapid increase in lung resistance in response to bradykinin (1 min), our measurements of LPA may underestimate the peak concentration.

In the en bloc preparation, µM concentrations of exogenous LPA caused increased carotid sinus nerve activity; and in the dual perfused in situ and naive in vivo preparations, it caused increased vagal activity and bronchoconstriction, respectively, both of which were abolished by carotid body denervation. Importantly, carotid sinus nerve activity was heightened in the en bloc carotid body preparation when plasma collected from asthmatic rats following OVA challenge was delivered into the preparation; an effect which was abrogated by TRPV1 and LPA receptor blockade and not demonstrated with plasma drawn from naive rats. Furthermore, blocking LPA signaling in the OVA model reduced bronchoconstriction (in anesthetized preparations) and ventilatory effects associated with respiratory difficulty (in conscious animals) following allergen provocation. Together, these data demonstrate that LPA present systemically following an allergen/bradykinin-provocation is likely a trigger of carotid body activity.

Approximately 70% of the en bloc carotid body's response to LPA was blocked by BrP-LPA (LPAr 1–4 antagonist) and Ki16425 (LPAr 1,3 and weak LPAr2 antagonist). We confirmed the presence of these LPA-specific receptors using RT-PCR: the carotid bodies contain LPAr 1, 3, 4, and 6 and the petrosal ganglia, containing cell bodies of carotid body chemosensory afferents, contain LPAr 3 and 6. Recent studies suggest LPA also activates TRPV1 both directly by binding to the C-terminus of TRPV1[35] and indirectly via LPAr activation and PKCε phosphorylation[36]. As TRPV1 is expressed in chemosensory

afferents[33] and TRPV1 antagonist AMG9810 blocks the LPA response of the en bloc preparation that remains after BrP-LPA or Ki16425 blockade, TRPV1 activation must also be part of the trigger mechanism. Interestingly, TRPV1 is activated by several other inflammatory mediators that have demonstrated effects on the carotid body, including IL-1, prostaglandins and TNFα[64]. These inflammatory mediators are also released by the lung during an acute asthmatic attack[2,4], but whether they act as additional triggers for carotid body-mediated bronchoconstriction remains to be determined.

Our data suggest new targets for pharmacological intervention in asthma. Although using an in vivo anesthetized ventilated preparation allows the ability to measure lung mechanics directly, it limits the ability to monitor asthmatic responses over extended periods due to the accumulating risk of lung injury. In a subsequent set of experiments, we delivered dual blockade following allergen challenge in order to test the efficacy of combined LPAr + TRPV1 inhibition as a method of therapy. Behavioral indices of airway resistance (Ti:Te) were reduced ~60% by combined blockade suggesting that the bulk of neurogenic bronchoconstriction in the ovalbumin rat model of asthma is dependent on the LPAr + TRPV1 pathway. Interestingly, our protocol used a randomized within-subject design allowing the effects of dual treatment, saline and vehicle to be tested and compared in the same rat. Remarkably, we found that the subset of animals receiving OVA + dual blockade on day 18 and receiving OVA + saline on day 21, had lower indices of airway resistance on day 21 than animals receiving OVA + saline on both days. The combined blockade of LPAr + TRPV1 as an asthmatic treatment is therefore a particular intriguing possibility because it reduces

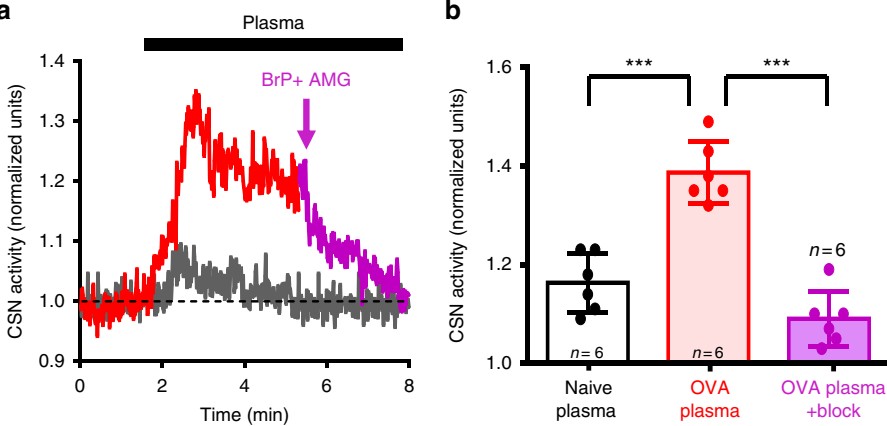

**Fig. 5** Plasma from ovalbumin-sensitized rats increases carotid body activity in LPA receptor-dependent manner. **a** Carotid sinus nerve activity from a naive en bloc carotid body preparation in response to plasma from naive (gray trace) and OVA-sensitized (red trace) rats. Application of dual LPAr and TRPV1 blockade (AMG9810, 10 μM + BrP-LPA, 1.5 μM) is indicated by the mauve arrow and subsequent trace. **b** Summary data of the effect of naive (open) and OVA (red) plasma as well as subsequent dual blockade (mauve). $F_{2,17}$ (one-way ANOVA) = 40.193, ***$p < 0.001$, mean ± sem. Holm–Šidák post hoc: OVA vs naive $p < 0.001$; OVA vs blockade $p < 0.001$

airway resistance even when administered following allergic induction and may have beneficial effects lasting several days.

A key question is whether our data may be translated to humans. The current study demonstrated neurogenic mechanisms of acute airway hyper-responsiveness/bronchoconstriction in the ovalbumin-sensitized Brown Norway rat model[42]. This model exhibits many features known to occur in human asthma, including goblet cell hyperplasia, smooth muscle proliferation as well as eosinophilic infiltration. However, the mechanism we describe has yet to be demonstrated in humans. Also, the duration or severity of disease may alter the relative contribution of this mechanism to pathology. For example, as the severity of lung inflammation increases, local effects of LPA within the lung and activation of TRPV1 receptors in the carotid body by non-LPA mediators of inflammation, may have a progressively more important role in causing acute bronchoconstriction. These same factors may differ between rats and humans.

For our in vivo experiments in which we examined the effects of LPA signaling via the carotid body reflex on lung function, we used bradykinin in place of methacholine. Although many studies have utilized methacholine in place of an allergen challenge in this model, we felt that methacholine activation of M3 muscarinic receptors on the airway smooth muscle might short-circuit any parasympathetic (cholinergic)-mediated bronchoconstriction. Bradykinin stimulates bronchopulmonary C-fibers, mucous secretion, edema and is present in BALF of humans and rats[5–14]. Moreover, bradykinin is released from mast cells and produced from kinninogen and kallidin present in the broncho-alveolar space and epithelial layer in response to allergen[10]. Thus, bradykinin provided a natural stimulus to trigger neural reflex circuits experimentally[5–14]. Still, the reflex pathway induced by bradykinin may not be identical to that induced during allergen challenge, and thus, it may have skewed the importance of the LPA-mediated lung–carotid body–lung reflex. We also note that responses to both bradykinin and capsaicin are enhanced in asthmatic lungs. While responses to bradykinin are carotid body dependent and specific to asthma, responses to capsaicin occur in naive lungs, likely via a C-fiber dependent, vagal–vagal-mediated reflex. This suggests that there may be multiple mechanisms of acute bronchoconstriction, with relative importance depending on the stimulus. The additive and/or multiplicative effects of these different bronchoconstricting pathways have yet to be fully resolved in inflammatory lung disease. Finally, we note that much of the efficacy of the dual blockade in relieving asthmatic symptoms was preserved when ovalbumin was used in conscious ovalbumin-sensitized animals. This suggests that the neuronal mechanism that we describe may also be affective in treating the chronic condition.

## Methods
**Animals**. Male Brown Norway (BN/Crl, p28–35, 80–150 g) and Sprague Dawley rats (p21–28, 50–80 g) rats were purchased from Charles River (QC). Where appropriate, the use of rat strain is described in the experimental procedures below. Experimental procedures were approved by the University of Calgary Animal Care and Use Committee. Animals were housed in pairs in a 12 h light/dark cycle with water and chow freely available.

**Chemicals and reagents**. Oleoyl-Lysophosphatidic acid (18:1 LPA), BrP-LPA, and AMG9810 were purchased from Cayman Chemical (Cayman, Ann Arbor, MI). D-(+)-sn-1-O-linoleoyl-glyceryl-3-phosphate (18:2 Linoleoyl LPA) was purchased from Echelon Biosciences (Salt Lake City, UT). 1-Palmitoyl-2-hydroxy-sn-glycero-3-phosphate (16:0 Lysophosphatidic acid) was purchased from Avanti Polar Lipids (Alabaster, AL). FURA-2AM was purchased from Invitrogen (Carlsbad, CA). Ovalbumin, pertussis toxin, aluminum hydroxide, $MgSO_4$, $NaH_2PO_4$, KCl, $NaHCO_3$, NaCl, glucose, sucrose, $CaCl_2$, Tween 80, dimethyl sulfoxide, pancuronium bromide, sodium cyanide, trypsin, Ham's F12, Ki16425, and ethanol were purchased from Sigma-Aldrich (Sigma-Aldrich, Oakville ON).

**Statistics and analysis**. All statistical analysis was performed in SigmaPlot Vs 13.0 (Systat Software San Jose, CA). All tests were two-sided and specific tests used are provided at the end of each specified experiment section. Normally distributed data were analyzed using parametric statistics and presented as mean ± sem; the Holm–Šidák post hoc test was used for pairwise multi-comparisons of normal data, unless otherwise stated. All other data were analyzed using non-parametric statistics and presented as median ± range.

**RT-PCR**. Carotid bodies, petrosal and superior cervical ganglia were collected from $n = 6$ Sprague Dawley rats (80–150 g, P21–28) and tissue were stored in RNALater (Sigma-Aldrich, St. Catherines, QC) at 4 °C until analysis. Total RNA extracted from isolated carotid bodies and superior cervical and petrosal ganglia was purified with the RNeasy Mini kit (Qiagen, Germantown, MD) per manufacturer's instructions. Total RNA (200 ng) from each sample was converted to single-stranded cDNA using the Quantiect RT-PCR (Qiagen, Germantown, MD) with random primers 10 μM. PCR amplification was carried out with a 20 μl reaction volume containing 1 μl of a cDNA, 7 μl ddH2O, primers at 1 μL each, and 10 μL PCR enzyme. PCR was performed under the following conditions: 95 °C for 3 min followed by 45 cycles of denaturation (95 °C for 30 s), annealing (60 °C for 30 s), and elongation (72 °C for 1 min), followed by 3 min at 72 °C before refrigeration (4 °C). The primer sequences for LPAr1, 2, 3, 4, 5, 6 and hypoxanthine phosphoribosyltransferase (HPRT) are presented in supplementary table 2. The amount of cDNA for each tissue was confirmed with a NanoDrop spectrophotometer (Thermo Scientific, Burlington Ontario). Changing the lane of the ladder identified which tissue was run on which gel. The PCR products were analyzed by electrophoresis using a 1% agarose gel and visualized under UV light with BioRad Image Lab 3.0 Software (Missisauga ON).

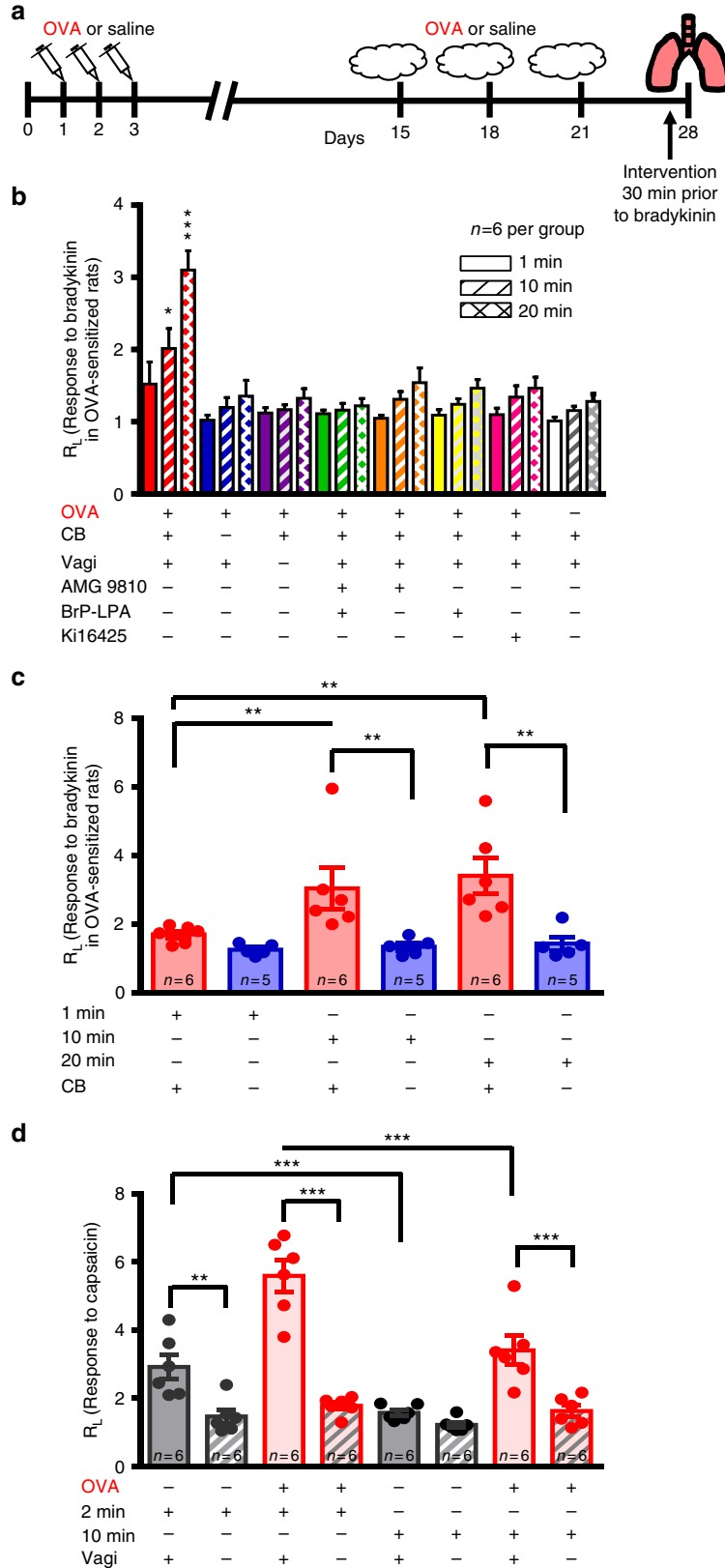

**Carotid body type I cell isolation and calcium imaging**. The procedures pertaining to calcium imaging were approved by the Wright State University Institutional Laboratory Animal Care and Use Committee. Sprague–Dawley rats (20-50g, P12-19) were anesthetized with isoflurane (3–5% in $O_2$), and carotid bodies were harvested and digested in 0.4 mg ml$^{-1}$ collagenase type I (Worthington Biochemical Corporation, Lakewood, NJ) and 0.2 mg ml$^{-1}$ trypsin type I (Sigma-

Aldrich, St. Catherines, QC) in DPBS enzyme solution with low $CaCl_2$ (86 μM) and $MgCl_2$ (350 μM), for 20 min at 37 °C followed by dissociation with forceps and incubated for an additional 7 min. The tissue was centrifuged (115 × g) for 3 min, supernatant removed and the pellet re-suspended in Ham's F12 (Sigma-Aldrich, St. Catherines, QC) supplemented with 10% heat inactivated fetal bovine serum (Biowest, San Marcos, TX). Cells were released by trituration with fire polished

**Fig. 6** Bradykinin-induced bronchoconstriction in ovalbumin-sensitized rats is dependent on the carotid body and LPA signaling. **a** OVA-sensitization protocol (see Methods, OVA Cohort 2) to test lung–carotid body–lung pathway. **b** OVA-sensitized and naive rats were exposed to nebulized saline (baseline) and three consecutive nebulizations of 0.4 mg bradykinin at 1 (solid), 10 (hatched), and 20 (crossed) min while measuring $R_L$. Bradykinin had group specific effects: See Methods, OVA Cohort 2; $F_{14,143}$ (two-way RM ANOVA: time × group) = 4.035, $p < 0.001$. Holm–Šidák post hoc: bradykinin caused a marked increase in $R_L$ in OVA-sensitized (red; **$p < 0.01$) but not naive rats (white; $p > 0.3$) rats; carotid body (CB) denervated (blue), vagi (VaG) denervated (purple), TRPV1 blockade (AMG9810, orange), LPAr blockade (BrP-LPA, yellow and Ki16425, fuscia) and dual TRPV1 and LPAr blockade (AMG9810 + BrP-LPA, green), abolished the effects of bradykinin compared to OVA (Holm–Šidák post hoc: *$p < 0.05$; ***$p < 0.001$). **c** Chronic CB denervation (blue; but not sham-treatment, red) also abolished bradykinin-induced bronchoconstriction in OVA-sensitized animals (see Methods OVA Cohort 3; $F_{2,32}$ (two-way RM ANOVA time × group) = 5.418, $p = 0.014$; Holm–Šidák post hoc: Intact greater than denervated, 10 and 20 min greater than 1 min in Intact, **$p < 0.01$). **d** Bronchoconstriction in response to capsaicin does not involve the carotid body (see Methods OVA Cohort 4). $R_L$ was measured 2 and 10 min after single capsaicin nebulization in OVA (red) and naive (gray) rats who had intact (solid) or resected vagi (hatched). All rats had denervated carotid bodies yet all, including naive animals, exhibited bronchoconstriction in response to capsaicin by 2 min ($F_{3,47}$ (two-way RM ANOVA treatment × group) = 8.435, $p < 0.001$). Holm–Šidák post hoc: capsaicin-induced bronchoconstriction was abolished by vagotomy: ***$p < 0.001$; different between 2 min and 10 min ***$p < 0.001$; **$p = 0.002$. In **b–d**, $R_L$ is normalized to that during initial saline nebulization. All data are presented as mean ± sem

---

silanised Pasteur pipettes (Sigma-Aldrich, St. Catherines, QC). Type I cells were plated on 15 mm round poly-D-lysine-coated (0.1 mg ml$^{-1}$) glass coverslips (Warner Instruments, Hamden, CT) and incubated at 37 °C in 5% $CO_2$, 10% $O_2$ ~2 h before use; cells were used for experiments within 8 h of isolation.

Type I cells were loaded with 5 μM FURA-2AM (Invitrogen, Carlsbad CA) in serum-free Ham's F12 nutrient media for 30 min at room temperature in humidified 5% $CO_2$, 10% $O_2$, before being transferred to FURA-2AM-free media in the same conditions for 20 min. Coverslips were placed in an RC-25F (Warner Instruments, Hamden, CT) 500 μl recording chamber at 34–36 °C. Image acquisition was controlled by Metafluor software (Molecular Devices, Sunnyvale CA), and cells were visualized using a Nikon TE2000-U inverted microscope with a CFI super fluor ×40 oil immersion objective. The FURA-2 loaded cells were excited by 50 ms exposures to 340/380 nm light using a Lambda 10-3 filter wheel every 5 s and emitted light was recorded at 510 nm using a Coolsnap HQ2 CCD camera (Photometrics, Tucson AZ).

Cells were continuously perfused with a standard HEPES buffered salt solution containing (in mM): 140 NaCl, 4.5 KCl, 2.5 $CaCl_2$, 1, $MgCl_2$, 11 glucose, 10 HEPES, adjusted to pH 7.57 with NaOH at room temperature to yield a pH of 7.4 at 37 °C. Solution containing LPA (5 μM, 300 s) or high potassium (20 mM, ≤70 s) was switched from independent reservoirs and superfused over the cover slips. The change in fluorescence ratio ($F_{340}/F_{380}$) from baseline to peak ($\Delta F_{340}/F_{380}$) was measured for each challenge. Significance was tested using one-way ANOVA of peak emission against baseline values.

**En bloc perfused carotid body preparation.** Sprague–Dawley rats (150–250 g) were anesthetized with isoflurane and then decapitated, the carotid bifurcation, including the carotid body, carotid sinus nerve, and superior cervical ganglion, was quickly removed and transferred to a beaker (100 ml) containing carbogen (95% $O_2$, 5% $CO_2$) equilibrated physiological saline (1 mM $MgSO_4$, 1.25 mM $NaH_2PO_4$, 4 mM KCl, 24 mM $NaHCO_3$, 115 mM NaCl, 10 mM glucose, 12 mM sucrose, and 2 mM $CaCl_2$). After ~20 min, the carotid bifurcation was transferred to a recording chamber with a built-in water-fed heating circuit (AR, custom made) and the common carotid artery was immediately cannulated for luminal perfusion with physiological saline (as above) with a peristaltic pump set at 15 ml min$^{-1}$ to maintain a constant pressure of 100 mmHg. The perfusate was equilibrated with computer-controlled gas mixtures of 100 Torr $PO_2$ and 35 Torr $PCO_2$ balanced with $N_2$ and recirculated throughout the experiments (yielding pH ~7.4) and heated to 37 ± 0.5 °C. The carotid sinus region was bisected, and the carotid sinus nerve was de-sheathed. Chemosensory discharge was recorded extracellularly from the whole de-sheathed carotid sinus nerve, hooked to a platinum electrode and lifted into a thin film of paraffin oil. A reference electrode was placed close to the carotid artery bifurcation. Nerve activity was monitored using a differential AC amplifier (model 1700, AM Systems), secondary amplifier (model AM502, Tektronix, Beaverton, OR), filtered (300-Hz low cut-off, 5-kHz high cut-off), displayed on an oscilloscope, rectified, integrated (200-ms time constant), and stored on a computer using an analog-to-digital data acquisition system (Digidata 1322A, Axon Instruments, Axoscope 9.0). Preparations were exposed to a brief hypoxic challenge (60 Torr $PO_2$) to determine viability. Preparations that failed to show a clear increase in activity during this challenge were discarded. After this challenge, preparations were left undisturbed for 30–45 min to stabilize before the experimental protocol was initiated. (1) LPA was infused at three separate concentrations (2.5, 5, 10 μM) $n = 6$ each for 18:1, 18:2, and 16:0 species, each. (2) TRPV1 blockade (AMG9810 dissolved in dH$_2$O 10 μM) was infused, 5 min later LPA (18:1) was infused (2.5, 5, 10 μM, $n = 6$). (3) LPAr blockade (BrP-LPA dissolved in DMSO 1.5 μM) was infused, 5 min later, LPA (18:1) was infused (2.5, 5, 10 μM, $n = 6$). (4) LPAr blockade (Ki16425 dissolved in DMSO 5 μM) was infused, 5 min later LPA (18:1) was infused (2.5, 5, 10 μM, $n = 6$). (5) LPAr blockade (BrP-LPA dissolved in DMSO 1.5 μM) was infused, 20 min later, LPA (18:1, 5 μM) was

infused, and subsequently TRPV1 blockade (AMG9810 dissolved in dH$_2$O 10 μM) was infused ($n = 6$). (6) LPAr blockade (Ki16425 dissolved in DMSO 5 μM) was infused, 2–5 min later, LPA (18:1, 5 μM) was infused, and subsequently TRPV1 blockade (AMG9810 dissolved in dH$_2$O 10 μM) was infused ($n = 6$). (7) 1 ml of plasma from naive Brown Norway rats after saline nebulization (as per model, below) was circulated through the preparation (~100 ml) for 10 min. (8) 1 mL of plasma drawn from ovalbumin-sensitized Brown Norway rats was drawn after ovalbumin challenge (below) and circulated through the preparation (~100 ml), 5 min later BrP-LPA (1.5 μM) + AMG9810 (10 μM) were co-infused. Neural traces were analyzed offline using custom software (written in VEE by R. J. A. W.). One minute of carotid sinus nerve activity during each condition was rectified, summed and expressed as integrated neural discharge. The neural responses for different conditions were normalized to the baseline (normoxic) condition. LPA species differences were analyzed with two-way repeated measures ANOVA (dose x species), all other data were analyzed with one-way ANOVA for each dose or between conditions.

**Dual perfused preparation.** Sprague–Dawley rats (80–150 g) were deeply anesthetized with isoflurane via inhalation. Then, rats were cooled in ≤4 °C physiological saline whilst maintaining isoflurane anesthesia. Once respiratory movement began to subside, the rat was decerebrated at midcollicular level, transected above the renal arteries and skinned. All tissue rostral to the decerebration and all remaining cortex dorsal to the colliculi were removed. Transection and decerebration were performed in ≤4 °C physiological saline containing 115 mM NaCl, 24 mM $NaHCO_3$, 4 mM KCl, 2 mM $CaCl_2$, 1 mM $MgSO_4$, 1.25 mM $NaH_2PO_4$, 10 mM glucose, and 12 mM sucrose, equilibrated with 95% $O_2$, 5% $CO_2$. Once dissection was complete, the preparation was placed in a supine position in a specially designed plexiglass chamber and secured with ear bars. The descending aorta was cannulated with a double-lumen catheter. One lumen of the catheter was connected to a peristaltic pump (Gilson Minipuls 3) and used to perfuse the descending aorta in a retrograde direction with perfusate at room temperature (20 °C) and equilibrated with 40 Torr $PCO_2$ in $O_2$ (central perfusion). The other lumen was attached to a pressure transducer and used to monitor perfusion pressure. After the cannulation procedure, the speed of the peristaltic pump was increased to elevate perfusion pressure to 90 mmHg over the first few minutes with use of a custom-built computer-controlled feedback system (built by R. J. A. W. written in VEE). The common carotid arteries were tied off above the clavicles and cannulated caudal to the carotid bifurcation. A separate peristaltic pump with two channels was used to perfuse the common carotid arteries at 18.5 ml min$^{-1}$ in order to maintain a perfusion pressure of ~90 mmHg. Up to this point in the dissection, central and peripheral perfusions were from the same tonometer. Independent perfusion of central (descending aorta) and peripheral (carotid arteries) circuits was then initiated by pulling fresh media from two different reservoirs of a custom-built tonometer. This custom-built system was designed to accommodate a common return while preventing mixing of perfusate once equilibration was achieved in the two reservoirs. Between the reservoir and the preparation, central and peripheral perfusate passed through a heat exchanger, bubble trapper, and 25 μm filter. After the initiation of independent perfusion, the central perfusate was equilibrated with 40 Torr $PCO_2$ in $O_2$, and the peripheral perfusate was equilibrated with 35 Torr $PCO_2$ and 100 Torr $PO_2$ in $N_2$. Of note, the decerebration transects the circle of Willis therefore preventing any arterial mixing of perfusates. Once stabilized and distinct breath-like movement was detected, the phrenic nerve was dissected free and attached to a suction electrode. The vagus nerves were dissected free and transected at the level of the clavicle, well below the nodose and vagal ganglia, and the right proximal vagus (descending from brainstem) was attached to a suction electrode. Neurograms were amplified (10,000x Phrenic, 20,000x Vagus; Differential AC Amplifier Model 1700, A-M Systems Inc., Carlsborg, WA, USA), filtered (low cut-off, 300 Hz, high cut-off, 5 kHz), rectified and

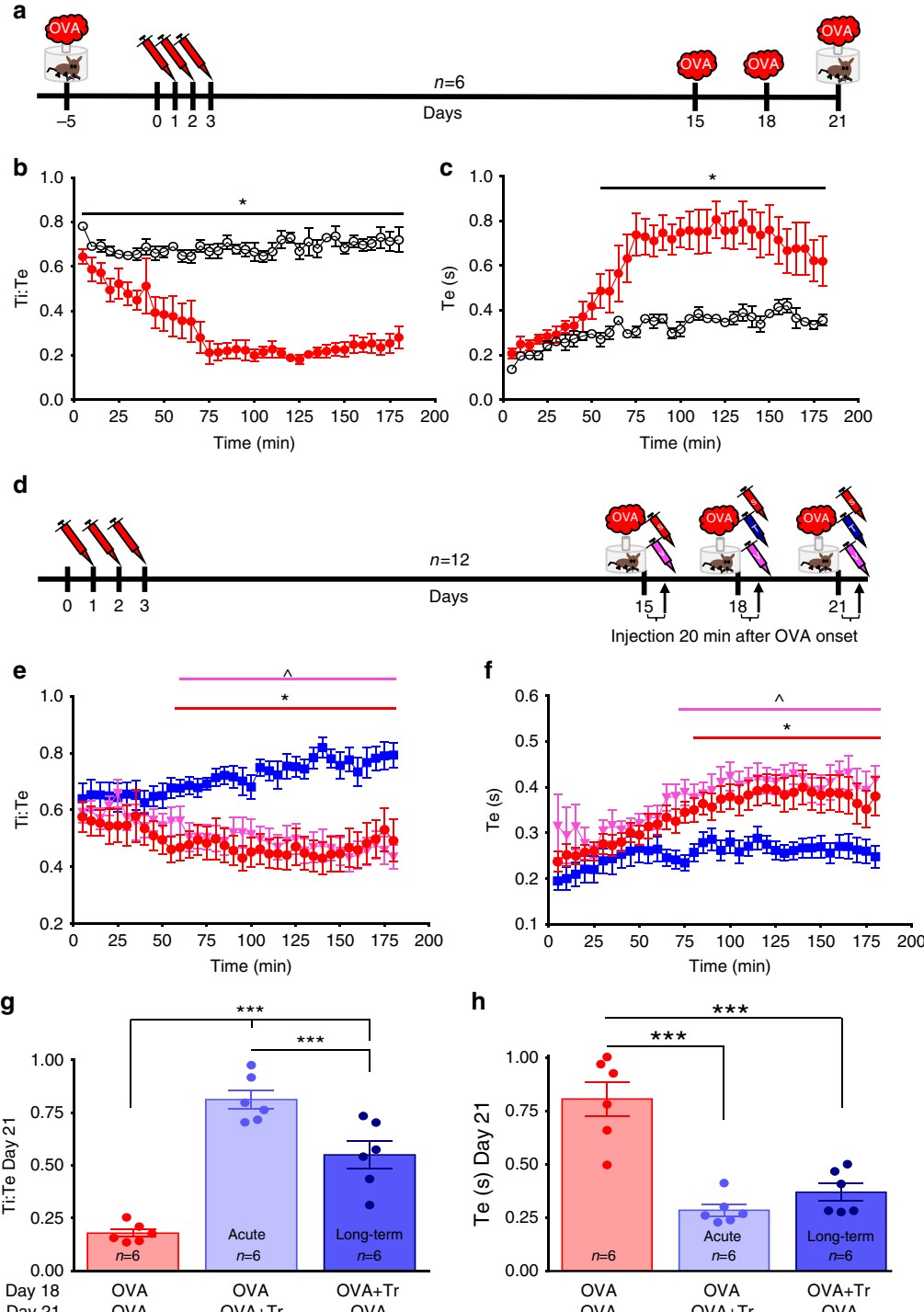

integrated (200-ms time constant, CWE moving averager, Ardmore, PA) and computer archived (Digidata 1322 A and Axoscope 9.0, Axon Instruments/Molecular Devices, Union City, CA, USA) at a sampling rate of 5 kHz, and analyzed off-line with custom software written in VEE (R.J.A.W.). Once carotid body activity was assessed with a hypoxic bout (50 Torr $O_2$, 40 Torr $CO_2$) normoxic conditions were re-established; any preparation which failed to demonstrate a hypoxic response was discarded. Upon recovery from hypoxia, the brainstem was made hypocapnic, in order to render the preparation apneic. Once apnea was achieved, LPA (18:1, 5 μM) was delivered into the line supplying the carotid body without recirculation. Ten minutes were given to record the presence or absence of a response, upon which normoxia was re-established to demonstrate the viability of the preparation. This experiment was completed in $n = 6$ carotid body intact and $n = 6$ carotid body denervated preparations. Variables were normalized to the initial normoxic condition. The difference between LPA and hypocapnic conditions was calculated for phrenic (nVT—neural tidal volume (amplitude, normalized units),

fR—frequency (bursts $min^{-1}$), nVE (neural minute ventilation, fRxnVT—normalized units) and vagal total activity (normalized units) and used for statistical analysis. Differences between intact and denervated preparations were compared with Mann Whitney Rank Test (fR, nVT, and nVE) or Student's unpaired two-sided $t$ test (Vagus).

**In vivo demonstration of LPA mediated bronchoconstriction.** To investigate the carotid body-mediated bronchoconstricting pathway in response to LPA, naive Brown Norway (160–200 g) rats were anesthetized with isoflurane (5%, balance $O_2$) and instrumented for surgery. The femoral artery and vein were cannulated for the measurement of arterial pressure, the infusion of intravenous anesthetic, alfaxan (~15 mg $kg^{-1}$ $min^{-1}$) delivered by syringe pump (Kent Scientific, Torrington CT) and the jugular vein was cannulated for the delivery of drugs and saline. The trachea was cannulated, and the rat was subsequently paralyzed with pancuronium

**Fig. 7** LPAr + TRPV1 blockade abates acute asthmatic respiratory distress in conscious rats. **a** OVA-sensitization protocol (see Methods, OVA Cohort 6). **b**, **c** Inspiratory:expiratory time decreased (Ti:Te; $F_{35,431}$(two-way RM ANOVA time × group) = 8.577, $p < 0.001$, Holm–Šidák post hoc: $p < 0.05$, * different between groups at indicated time), expiratory time increased (Te; $F_{35,431}$ (two-way RM ANOVA time × group) = 3.948, $p < 0.001$, Holm–Šidák post hoc: $p < 0.05$, * different between groups at indicated time) in response to acute OVA provocation following OVA sensitization, confirming these parameters as indices of acute asthmatic respiratory distress in conscious animals. **d** 21-Day sensitization protocol to test dual LPAr + TRPV1 blockade on respiratory distress (see Methods OVA Cohort 7); LPAr + TRPV1 blockade (T), vehicle (V), or saline (S; randomized) were delivered 20 min after the onset of OVA (as indicated by the arrows). **e** Decrease in Ti:Te and **f** increase in Te caused by allergen provocation are rescued by dual blockade (blue). Ti:Te: $F_{70,1293}$ (two-way RM ANOVA group × time = 3.169, $p < 0.001$; Te: $F_{35, 385}$ (two-way RM ANOVA time) = 10.590, $p < 0.001$, $F_{2, 35}$ (two-way RM ANOVA group) = 7.393, $p = 0.004$). Holm–Šidák post hoc: dual block is significantly different from OVA-sensitized saline injected (red circles)* and vehicle injected (pale red triangles)^ groups, at indicated time points, $p < 0.05$. **g** The peak Ti:Te responses recorded 120 min after OVA exposure on day 21 in animals never having received dual blockade (red—from **b**), having dual blockade on day 21 and recorded on day 21 (acute treatment, light blue, from **e**), or having dual blockade on day 18 and recorded on day 21 (Long term treatment, dark blue, from **e**; $F_{2,15}$ (one-way ANOVA group) = 45.805, $p < 0.001$). **h** The peak Te (120 min) response recorded on day 21 following OVA exposure; groups as per **g**. Dual antagonist injection on days 18 or 21 reduced respiratory indices of acute bronchoconstriction; and remarkably, dual antagonist injection on day 18 also had beneficial effects three days later, on day 21, without a subsequent dual antagonist injection ($F_{2,15}$ one-way ANOVA group = 25.906, $p < 0.001$). Holm–Šidák post hoc: difference between indicated groups ***$p < 0.001$. Data are presented as mean ± sem

bromide (1 mg kg$^{-1}$, i.a., dissolved in 0.9% saline) and the animal attached to the Flexivent respirator system (SCIREQ, Montreal QC) for ventilation and measurement of airway resistance. Upon stabilization to the ventilator and intervention, single oscillator maneuvers (Snapshot 90) were repeated five times during saline (control condition), injection of 5 μM LPA bolus into the jugular vein, and finally an injection of 6 μM bolus sodium cyanide (i.v., an independent test of carotid body function[41]) at least 10 min were given between challenges. This experiment was repeated in $n = 6$ rats with intact carotid sinus nerves and $n = 6$ bilateral carotid sinus nerve denervated rats.

In order to demonstrate that C-fiber mediated vagal–vagal reflexes remain intact in carotid body denervated rats, naive Brown Norway carotid body denervated rats (160–200 g) with vagus intact ($n = 6$) or denervated ($n = 6$) were exposed to aerosolized capsaicin (100 breaths, 50 μM) and lung resistance was measured at 2 and 10 min[65]. The average of the five single oscillator maneuvers was taken to calculate total lung resistance ($R_L$) for each time point expressed as normalized units (normalized to the baseline saline condition). Data were analyzed using Student's unpaired two-sided $t$ test for each condition for LPA or NaCN injection. Capsaicin data were analyzed with two-way repeated measures and one-way repeated measures ANOVA.

**Asthmatic model.** Brown Norway rats (80–120 g) were sensitized to ovalbumin (1 mg) with pertussis toxin (0.5 ng) and aluminum hydroxide as adjuvant (0.15 g) dissolved in saline (1 ml i.p.; OVA sensitized), or saline (1 ml i.p.; Naive) for three consecutive days (Days 1, 2, and 3) and challenged with aerosolized 5% ovalbumin (OVA, dissolved in 0.9% saline) or saline (Naive) on days 15, 18, and 21 for 10 min. All anaesthetized experiments conducted on OVA and Naive rats were conducted ~7 days (day 28) following the last aerosol exposure. Differences in timeline are indicated for specific experiments below.

**Broncho-alveolar lavage fluid.** Bronchoalveolar lavage fluid was collected from $n = 7$ OVA and $n = 7$ Naive rats following challenge with bradykinin (below). With the upper trachea cannulated, lungs were lavaged (10 ml per lavage) with saline (0.9%). Cells in bronchoalveolar were sedimented by centrifugation (20 min at 4500 × $g$, 4 °C) and resuspended in phosphate buffered saline. 100 ml of bronchoalveolar lavage fluid was centrifuged (Shandon Cytospin 4 cytocentrifuge, Thermo Scientific, Waltham, MA, 6 min at 4500 × $g$) and cells collected on non-coated glass slides, fixed in 95% ethanol and stained with hematoxylin and eosin. Total leukocytes were determined by hemacytometer counting. Differentiation of 200 cells was completed according to standard morphologic criteria[66]. Samples were compared with Student's unpaired two-sided $t$ test.

**Lung histology, immunohistochemistry, and gene expression.** Following exposure to bradykinin (below) both lungs were inflated with formalin (10–15 ml, ~30 mm Hg) in $n = 7$ OVA and $n = 7$ Naive rats, rapidly excised then fixed in formalin. The left lung was hemisected at the level of the bronchus[67], embedded in paraffin and subsequently 4 μm thick sections cut and de-paraffinized. To determine goblet cell metaplasia, sections were stained with periodic acid and schiffs reagent and counter-stained with hematoxylin. The numbers of goblet cells were expressed as cells per circumference of bronchial epithelium measured using Image J (NIH, Bethesda ML). Immunohistochemistry was performed for α-smooth muscle actin (αSMA; primary antibody: mouse anti-αSMA with biotin tag Thermofisher Cat# MS113-01, 1:143 in 2% normal goat serum; secondary antibody: goat anti-mouse IgG with biotin tag Vector Cat# BA9200, 1:200 in 2% normal goat serum) using DAB as chromogen. Smooth muscle hyperplasia was quantified as the intensity of stain per area of section using Image J (NIH, Bethesda ML). Total inflammation score was evaluated by a blinded observer (MMK) using a semi-quantitative scoring system to evaluate the fraction of the airway that was occupied

by inflammatory cell infiltrates: 4 = robust inflammation (more than 50% of airway circumference surrounded by inflammatory cell infiltrates); 3 = moderate inflammation (25–50% of airway circumference surrounded by inflammatory cell infiltrates); 2 = mild inflammation (10–25% of airway circumference surrounded by inflammatory cell infiltrates); 1 = minimal inflammation (<10% of airway circumference surrounded by inflammatory cell infiltrates) and a score of 0 = no inflammatory cell infiltrates; for lung parenchyma the same system was used but in reference to the percent of alveoli involved by inflammation.

Gene expression of IL4 and eotaxin (CCL11) were probed with SYBR green qPCR in relation to β-actin. Total RNA extracted from left lungs was purified with the RNeasy Mini kit (Qiagen, Germantown, MD) per manufacturer's instructions. The amount of cDNA for each tissue was confirmed with a NanoDrop spectrophotometer (Thermo Scientific, Burlington Ontario). Total RNA (1000 ng) from each sample was converted to single-stranded cDNA using the Quantiect RT-PCR kit (Qiagen, Germantown, MD) with 10 μM of random primers. PCR amplification was carried with a 20 μl reaction volume containing 1 μl of a cDNA, 7 μl ddH$_2$O, primers at 1 μL each, and 10 μL SYBR green (QuantiNova Qiagen, Germantown, MD). PCR was performed under the following conditions: 95 °C for 3 min followed by 45 cycles of denaturation (95 °C for 30 s), annealing (60 °C for 30 s), and elongation (72 °C for 1 min), followed by 3 min at 72 °C before refrigeration (4 °C) with melt curves produced for all samples; samples for each rat and gene were run in triplicate with the Eppendorf Mastercycler (Eppendorf, Mississauga ON). The primer sequences for IL4, CCL11 (eotaxin) and β-actin are presented in supplementary table 2. Data are expressed as cycles threshold and $2^{\Delta CTT}$ was calculated by standard methods[68]. Data were compared with Student's unpaired two-sided $t$-test or Mann Whitney Rank Sum Test (inflammation score).

**LPA concentration.** Arterial blood samples (0.7 ml) were drawn prior to and following bradykinin challenge (~20 min after initial bradykinin exposure) in anesthetized experiments. Blood samples were spun for 20 min at 4000 × $g$ in heparinized (5 IU) tubes at room temperature, plasma was drawn and snap frozen and stored at −80 °C until analysis.

**ELISA.** Plasma samples were tested for LPA concentration by the K-2800S ELISA plate (LPath Inc., San Diego, CA; Echelon Biosciences, Salt Lake City, UT) in strict accordance with manufacturers specifications. Samples were run in duplicate as per manufacturers recommendations and coefficient of variation calculated. Samples were analyzed by two-way repeated measures ANOVA.

In order to verify the standard curve under experimentally appropriate conditions (1) the curve was compared with a plasma sample spiked with known LPA 18:1 concentrations; (2) plasma preparation with EDTA and Heparin were compared; and (3) venous vs arterial samples were compared (supplementary Fig 5). Data were analyzed with Pearson correlation (standard curves) and Students' independent $t$-test.

**Liquid chromatography–mass spectrometry.** LC/MS analysis was conducted by The University of Victoria GENOME Canada Protein Centre.

A mix containing 5 nmol each of four reference standards (16:0, 18:1, 18:0 lysoPAs, and 16:0 cyclic LPA) was dissolved in 1 mL of methanol, as standard solution. 100 μL of standards was mixed with 100 μL of the internal standard solution and used as reference standards. 50 μL of rat plasma was mixed with 100 μL of an internal standard solution [lysoPA(17:0), 0.1 nmol mL$^{-1}$ in methanol], vortexed and added to an additional 350 μL of methanol. After centrifugal clarification at 21,000 × $g$, 10 °C for 15 min, the supernatants dried in a N$_2$ evaporator. The residue was dissolved in 200 μL of methanol and 10 μL was used for LC/MS analysis.

LC/MS runs were performed with the use of a C18 UPLC column for chromatographic separation on an Agilent UHPLC system coupled to a Sciex 4000

QTRAP mass spectrometer equipped with an electrospray ion (ESI) source, operated in the multiple-reaction monitoring (MRM) mode with negative-ion (−) detection. The mobile phase was 5 mM NH4Ac in water (A) and acetonitrile–isopropanol (1:1) as the mobile phase for gradient elution (45–100% B in 10 min at 0.4 mL min$^{-1}$ and at 60 °C). The targets for LC/MS detection were: LysoPA(14:0)D_381.5/153, LysoPA(16:0)_409.5/153, LysoPA(16:1)D_407.5/153, LysoPA(18:0)D_437.5/153, LysoPA(18:1)_435.5/153, LysoPA(18:2)_433.5/153, LysoPA(18:3)_431.5/153, LysoPA(20:2)_461.5/153, LysoPA(20:3)D_459.5/153, LysoPA(20:4)D_457.5/153, LysoPA(20:5)D_455.5/153, LysoPA(22:4)D_485.5/153, LysoPA(22:5)D_483.5/153, LysoPA(22:6)D_481.5/153, CyclicPA(16:0)_391.3/255.2, CyclicPA(16:1)D_393.3/257.2, CyclicPA(18:0)D_419.3/283.2, CyclicPA(18:1)D_417.3/281.2, CyclicPA(18:2)D_415.3/279.2, CyclicPA(18:3)D_413.3/277.2, CyclicPA(20:4)D_439.4/303.2, CyclicPA(22:6)D_463.4/327.2. LPA molecular species were selected with the closest possible approximation with available LPA standards for concentration calculation. Comparison of LC/MS to ELISA (of the 10 samples measured by both techniques) were compared with Pearson correlation.

**Lung mechanics and respiratory distress**. Seven cohorts of OVA sensitized Brown Norway rats were used for these experiments (see above).

OVA Cohort 1 ($n = 26$): To determine the effect of bradykinin and OVA-sensitization on airway resistance (R$_L$) OVA-sensitization (day 28) and naive rats were anesthetized with isoflurane (5%, balance O$_2$) and instrumented for surgery. The femoral artery and vein were cannulated for the measurement of arterial pressure, the infusion of intravenous anesthetic, alfaxan (15 mg kg$^{-1}$ min$^{-1}$ delivered by syringe pump; Kent Scientific, Torrington CT) and the jugular vein was cannulated for the delivery of saline (and drugs in several of the cohorts described below). The trachea was cannulated and the rat was subsequently paralyzed with pancuronium bromide (1 mg kg$^{-1}$, i.a., dissolved in 0.9% saline) and the animal attached to the Flexivent respirator system (SCIREQ, Montreal QC) for ventilation and measurement of airway mechanics. Bradykinin (0.4 mg) was nebulized for 30 breaths at 1, 10, and 20 min following an initial (saline) baseline challenge. Single oscillator maneuvers (Snapshot 90) were repeated five times during baseline challenge and each bradykinin inhalation and the average of the 5 maneuvers was taken to calculate total lung resistance (R$_L$) for each time point. Responses to bradykinin was normalized to the initial saline baseline challenge. Data were analyzed with two-way repeated measures ANOVA (time × group) and are presented in Fig. 4a, g.

OVA Cohort 2 ($n = 48$): To investigate the carotid body–vagal–lung bronchoconstricting pathway, naive and OVA-sensitized rats were instrumented and attached to the Flexivent as in Cohort 1. OVA-sensitized rats were randomly separated into seven different groups: no intervention (OVA), vagotomy (VaG), carotid body denervation (CB), LPAr blockade (BrP-LPA; 3 mg kg$^{-1}$ i.v., dissolved in dimethyl sulfoxide), LPAr blockade (Ki16425; 5 mg kg$^{-1}$ i.v., dissolved in dimethyl sulfoxide), TRPV1 Blockade (AMG9810; 10 µM kg$^{-1}$ i.v., 10x solution dissolved in 0.6 ml Tween 80 and 0.4 ml 100% ethanol and diluted in ddH$_2$0), LPAr + TRPV1 blockade (i.v.). Once surgical procedure and intervention (antagonist injection or nerve dissection) were complete, rats were allowed to stabilize for 30 min while being ventilated. Airway mechanics were measured in response to bradykinin nebulizations (as in Cohort 1). Heart rate was attained from the blood pressure waveform, SaO$_2$ was monitored from a pulse oximeter (Kent Scientific, Torrington CT) and blood gases were analyzed from 0.1 ml arterial samples drawn before and after bradykinin with a blood gas analyzer (Element POC, Heska Barrie, ON). Data were analyzed using two-way repeated measures ANOVA (group × time) and are presented in Fig. 6a, b and Supplementary Fig. 7.

OVA Cohort 3 ($n = 11$): To ensure effects of carotid body denervation were persistent, we tested the effects of bradykinin 4–5 days after carotid body denervation in OVA sensitized animals. Bilateral carotid body denervation was performed under ketamine/xylazine (100 mg kg$^{-1}$ IM) anesthetic. In sham counterparts, carotid bodies were identified but left intact. Both groups of animals received buprenorphine (0.05 mg kg$^{-1}$ SQ) post operatively. Airway mechanics were measured in response to bradykinin nebulizations (as in Cohort 1). Data were analyzed with two-way repeated measures ANOVA (group × time) and are presented in Fig. 6c.

OVA Cohort 4 ($n = 12$): To test if the bradykinin-mediated asthmatic bronchoconstriction involves a distinct mechanism to the C-fiber mediated vagal–vagal reflex, all animals were bilaterally carotid body denervated to eliminate the lung–carotid body–lung reflex. OVA sensitized rats were instrumented to the Flexivent (see Cohort 1) and carotid body denervated. Six rats were also vagotomised to evaluate the contribution of the C-fiber mediated vagal–vagal pathway. Upon stabilization, the C-fiber mediated vagal–vagal pathway was activated with aerosolized capsaicin (100 breaths, 50 µM) and airway mechanics measured at 2 and 10 min[65]. Data were analyzed with two-way repeated measures ANOVA (group × time) and are presented in Fig. 6d and supplementary Fig. 8a, b.

Cohort 5 ($n = 12$): To test if LPA has local bronchoconstricting effects, rats were instrumented to the Flexivent (see Cohort 1) and either bilaterally carotid body denervated or sham exposed. Upon stabilization, LPA (5 µM) was nebulized for 100 breaths and R$_L$ measured at 1 and 30 min post inhalation[58] as above. Data were analyzed with two-way repeated measures ANOVA (group × time) and are presented in supplementary Fig. 8c.

OVA Cohort 6 ($n = 6$): To characterize breathing in the asthmatic model, rats were studied in a plethysmograph. In this cohort, prior to sensitization, rats were exposed to OVA aerosol for 10 min and then placed in a plethysmograph (Buxco, DSI systems Minneapolis, MN) to obtain baseline measurements. On day 21, following their final aerosol exposure, plethysmography was repeated to determine effects of OVA sensitization. Increases in expiratory time (Te) and decreases in inspiratory time: expiratory time ratio (Ti:Te) were evaluated as indices of respiratory difficulty/airway constriction. Data were averaged in 5-min bins and compared using two-way repeated measures ANOVA (time × group) and are presented in Fig. 7a–c.

OVA Cohort 7 ($n = 12$): To demonstrate therapeutic effectiveness of the antagonist intervention following allergen provocation, a subset of $n = 12$ OVA rats underwent separate plethysmograph experiments. Rats were OVA-sensitized as described above, but on day 15, following OVA exposure, rats were treated with saline or vehicle; on day 18 and 21, aerosolized ovalbumin was delivered for 10 min, followed 10 min later by saline, vehicle, or LPA receptor blocking cocktail delivered i.p. The LPA receptor blocking cocktail contained 3 mg kg$^{-1}$ BrP-LPA to block LPA-specific receptors (dissolved in dimethyl sulfoxide) and 10 µM kg$^{-1}$ AMG9810 to block TRPV1 (10× solution dissolved in 0.6 ml Tween 80 and 0.4 ml 100% ethanol and diluted in ddH$_2$O). Rats were then placed in a plethysmograph (Buxco, DSI systems Minneapolis, MN; total time after beginning of aerosolization = 20 min) and Ti:Te and Te were measured as above for 3 h in order to attain the early and late-onset of asthmatic responses[69]. Five-minute averages of each variable were calculated. Data was analyzed using a two-way repeated measures ANOVA (time × group) and are presented in Fig. 7d–h.

**Code availability**. Custom written computer code can be run with VEE runtime 9.32 and is available from the corresponding author upon reasonable request.

## Data availability

The datasets generated and/or analyzed during the current study are available from: https://figshare.com/projects/Preventing_acute_allergen-induced_asthmatic_symptoms_by_targeting_lysophosphatidic_acid_receptors_in_the_carotid_body/37004.

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

## Acknowledgements

The authors would like to thank Elaine de Heuvel and Frank Visser for their expertise and invaluable assistance in conducting experiments. This study was supported by the Canadian Institute for Health Research and The Lung Association. R.J.A.W. is an Alberta Innovates Health Solutions Senior Scholar. N.G.J. received postdoctoral salary support for this project from the Canadian Allergy, Asthma and Immunology Foundation and Alberta Innovates Health Solutions.

## Author contributions

N.G.J., F.H.Y.G., and R.J.A.W. conceived and designed the research; N.G.J., A.R., N.O.B., L.T.-L. and R.L.P. performed the experiments; N.G.J., A.R., N.O.B., M.M.K., R.L.P., C.N. W., and R.J.A.W. analyzed the data; N.G.J., A.R., M.M.K., and R.J.A.W. interpreted the results of the experiments; N.G.J., A.R., and R.J.A.W. prepared the figures; N.G.J., and R. J.A.W. drafted the manuscript; N.G.J., and R.J.A.W. edited and revised the manuscript; All authors approved the final version of the manuscript.

## Additional information

**Competing interests:** N.G.J., A.R., and R.J.A.W. declare the following competing interests. U.S. Patent Application No. PCT/CA2018/000145, Status: provisional patent; "Method to Abate Acute Airway Hypersensitivity and Asthma Attacks". Purpose: for the use of TRPV1 and LPAr blockade as a treatment for respiratory distress associated with acute asthmatic attack. The remaining authors declare no competing interests.

