## [Peer Review File · Nature Communications]

Reviewers' comments:

Reviewer #1 (Remarks to the Author):

Jendzowsky et al studied the role of LPA in airway hyper-reactivity in a rat model of allergic airway inflammation. LPA is a pleiotropic lipid mediator and when produced extracellularly can bind G-protein coupled receptors on target cells. Although elevated lung LPA levels have been detected in allergic conditions, the function of this molecule is not known with certainty. One of the challenges in this area is that LPA is difficult to measure, and there are limited reagents to block its production or receptors.

The authors used sophisticated techniques and demonstrate that LPA is involved in a lung-carotid body reflex circuit leading to bronchoconstriction. Although LPA has been shown to activate TRPV1 previously, this is a novel demonstration of how LPA/TRPV1 might function in a physiologically relevant neural circuit in the lung. The role of the carotid body in airway responses in asthma is controversial, but the results are internally consistent and supported by complementary approaches. If true, the pathway described in this report has strong potential translational significance, since it would suggest that inhibiting LPA production / signaling is a new approach in asthma.

I have a few concerns that dampened my enthusiasm for the manuscript in its present form. Roughly in order of importance, these include the following.

1. The concentrations of LPA used in different preparations and detected in rat plasma are in the micromolar range, which seems very high. Most prior studies reported that serum or lung LPA levels are in the nanomolar range (e.g. PMID 18583620, 24872406, 17359381, and PMC5521143). The use of ELISA to measure LPA is not in widespread practice, and many investigators prefer direct measurement using LC/MS. The authors need to justify their choice of ELISA and put their results in context with prior literature. It would be reassuring if they could confirm the high values of LPA observed using LC/MS, to complement the ELISA approach.
2. Most results seem to come from small numbers of animals per group (e.g. n=6). Please confirm that results have been confirmed in additional cohorts and replicates, which will help ensure reproducibility of results.
3. In Figure 2 (panels h and i), it would be helpful to report the baseline and LPA-induced resistance (e.g. in the text), so the reader can appreciate the magnitude of the LPA effect in this model.
4. In Figure 3, please clarify when the antagonists were administered in relation to Ova challenges. It would be helpful to indicated this in the figure timeline, similar to Figure 4, for consistency.
5. I recognize that the main focus of this paper is AHR and not inflammation per se. Are there any readouts of inflammation that can be shown in Figure 3?
6. Since BrP-LPA acts as an autotaxin inhibitor, as well as LPAR antagonist, it could be reducing LPA generation in some models. This needs to be addressed.

Reviewer #2 (Remarks to the Author):

This paper uses the ovalbumin sensitized and challenged Brown Norway rat as a model of allergic asthma. As with several other published studies, the data shows that these animals have an augmented vagal reflex response. In the present study the reflex response specifically to nebulized bradykinin is augmented in the allergically inflamed animals. Bradykinin is a strong activator of vagal C-fibers in the lungs of all laboratory animals. The increased response to bradykinin is therefore consistent with the findings of others that show vagal C-fiber responsiveness is nonselectively enhanced in this model e.g., Kuo and Lai have found that the C-fibers in the lungs are hyperresponsive to adenosine, capsaicin, 5-HT agonists in this same model (J. Appl. Physiol, 2008). There are two findings that are novel and nicely supported; that LPA receptors are

expressed by glomus cells and by petrosal neurons; and that LPA can effectively stimulate CB activity.

General points

1. According to the authors a motivating factor of this study is to explain why carotid body resection is effective in human asthma. This seems somewhat disingenuous given the intense skepticism about any beneficial effect of carotid body resection on asthma. In fact, a position paper was published about this in JACI in August 1986. The authors point out that a review of the literature shows that when the patients were properly diagnosed and followed-up over time, there was no benefit of carotid body resection on asthma. Most of the studies are on unilateral resection, but the same skepticism is found in modern literature with the more dangerous bilateral resections. Therefore, I think it is misleading to presume the present data speaks to mechanisms underlying the efficacy of carotid body resection in human asthma.

If the paper would stick closer to the interesting basic science presented, and not so much on trying to prove asthma mechanisms, it would be an easier paper to follow. Bradykinin nebulization is referred to as "asthmatic provocation" in several places. The second sentence of the discussion referred to blocking LPA signaling as ameliorating acute asthmatic bronchoconstriction..." when I think "asthmatic bronchoconstriction" here refers to nebulized bradykinin-induced cholinergic reflex activity. The title refers to acute asthma attacks, when no such thing is studied, etc..

3. A novel and very interesting aspect of this study is that it argues that the enhanced parasympathetic reflex response to nebulized bradykinin in allergically inflamed rat airways is abolished by acute denervation of the carotid bodies. This would mean that activity in the carotid sinus nerve is somehow required for the classic and well studied airway vagal C-fiber induced vagal parasympathetic bronchoconstrictor reflex.

This would be very important, but also very peculiar. How do the authors imagine such a phenomenon occurring? The data presented do not make for a compelling case. To draw such a bold conclusion, more work is required. Is it possible that there is some artefactual effect of acutely ablating the carotid sinus nerves such that all vagal reflexes are blocked? Acutely severing inputs to the brainstem can have transient and profound effects. For example, bilateral severing of the vagus nerves in rodents can lead acutely to a complete loss of respiratory control and death. The hypothesis at hand is that the carotid bodies are needed for bradykinin-induced reflexes in allergically inflamed lungs. Might this be better addressed with a model whereby the carotid sinus nerves are cut in a survival surgery procedure. Then after the animals have had sufficient time to recover (~a week) one can determine if augmented vagal cholinergic airway reflexes are still blocked?

4. In the present study bradykinin, surprisingly, only evoked a reflex bronchoconstriction in allergically inflamed lungs? Does acute carotid sinus nerve ablation also prevent vagal airway reflexes in naïve animals. In other words, does acute severing of the carotid bodies abolish airway reflexes evoked by a stimulus that does not require airway inflammation to manifest itself?

5. The hypothesis here is that OVA leads to (a very modest) increase in plasma LPA and this is a necessary precondition for bradykinin applied to the lungs to evoke a bronchoconstrictor reflex (a reflex that somehow is dependent on activity in the carotid bodies).

This conclusion is based only on the observation that BrP-LPA a potent LPA-receptor antagonist (with other actions such as phospholipase and phosphodiesterase inhibitory activity) abolished the ability of bradykinin to evoke a cholinergic reflex. Does BRP-LPA cross the BBB where it may interfere with neurotransmission? Perhaps phospholipase is needed for bradykinin to activate C-fiber terminals? Although the MS states bradykinin increases serum LPA in the OVA rats, the data in figure 3vi do not support this assertion. The baseline serum LPA is slightly elevated by OVA, but bradykinin does not seem to further enhance the LPA concentration. Is the LPA increased by the repetitive allergen challenge somehow required for bradykinin to activate the airway C-fibers? Or is the argument that bradykinin effectively activated the airway C-fibers but can only lead to a cholinergic reflex if LPA in the serum facilitates carotid body activity. This is very confusing to me—it would help if the authors stated their specific hypothesis and then presented the data in its favor. At a minimum one would wonder if more selective LPA_r antagonists that are not enzyme inhibitors mimic the effect of BrP-LPA.

Since LPA receptors are expressed by the petrosal neurons, it would seem likely that they are also expressed nodose neurons. The vagal C-fibers terminate in the epithelium therefore these nerve endings are likely to come in contact with the largest concentration of the LPA—presuming, as stated, that OVA is stimulating LPA release from airway epithelium. Does nebulized LPA into the lungs evoke the reflex, as you argue that it will when infused to the CBs? If so is this airway reflex also blocked by lesioning the carotid sinus nerves. Can infused LPA into the lungs enhance the bradykinin evoked reflex or does it indeed need to reach the CBs?

We would like to thank the Reviewers for their time and consideration of our manuscript. We feel the reviewers have raised a number of important point which we have addressed below.

Reviewer #1 comments:

1. The concentrations of LPA used in different preparations and detected in rat plasma are in the micromolar range, which seems very high. Most prior studies reported that serum or lung LPA levels are in the nanomolar range (e.g. PMID 18583620, 24872406, 17359381, and PMC5521143). The use of ELISA to measure LPA is not in widespread practice, and many investigators prefer direct measurement using LC/MS. The authors need to justify their choice of ELISA and put their results in context with prior literature. It would be reassuring if they could confirm the high values of LPA observed using LC/MS, to complement the ELISA approach.

In light of this comment we have performed an extensive literature search of methods used to measure LPA and the concentration determined. Please see attached excel file.

LPA extended into the μM range in two of the four references cited by Reviewer 1 (see yellow highlighted in spreadsheet). In all we found 59 papers that measured LPA; 38 found LPA to be in the μM range. 30 used LC/MS, 17 of which found LPA in the μM range -- similar to that which we detected with ELISA -- with the latest in 2017. 10 studies focused on lung disease; 3 of which found LPA extended into the μM range in blood, 1 which found LPA extended into the μM range in BALF, 2 of these studies used LC/MS. We attempted to the online portal, but the PDF conversion made readability difficult. To access an MS Excel version of this file please see:

<https://www.dropbox.com/s/m92rt4vwwj89p29/LPA%20concentration%20meta%20analyses.xlsx?dl=0>

We choose to use ELISA because lipids are highly malleable and LC/MS involves multiple steps including a lipid extraction process which can be problematic. For example, alkenyl-acyl lipids in plasma can break down into LPA during the extraction process, which is still being optimized (Li et al. Int J Mol Sci 2014, 15, 10492-10507).

In recognizing Reviewer 1's concern however, we did additional validation experiments to confirm that asthma causes a physiological increase in plasma LPA sufficient to stimulate the carotid body:

A) We developed a new bio-assay to test if the elevated LPA in plasma from asthmatic animals is sufficient to increase carotid body activity.

We took plasma from asthmatic rats 3hr after OVA challenge and perfused it through the *en bloc* carotid body preparation (1 ml of plasma in 100ml of perfusate; see revised manuscript for details, methods and Fig 4). Plasma from asthmatic rats increased carotid sinus nerve (CSN) activity by ~45% (plasma from naïve rats increased CSN activity by less than ~15%). Moreover, LPA_r and TRPV1 antagonists abolished this increase. This experiment demonstrates the increase in plasma LPA translates to physiological effects on the carotid body.

B) We further interrogated the accuracy of ELISA (Supplementary Fig. 4 in revised manuscript). (a) We compared LPA calibration curves with and without plasma; plasma had no bearing on the outcome. (b) We compared LPA concentration in venous and arterial asthmatic plasma; there was no significant difference. (c) We compared EDTA and heparin processing of samples; there was no significant difference. (d) We evaluated the sensitivity of *en bloc* carotid body activity to additional species of LPA (16:0 and 18:2) that are more common in blood than the LPA we used previously (18:1); carotid body activity is sensitive to all three LPA species (Fig 1 e).

2. Most results seem to come from small numbers of animals per group (e.g. n=6). Please confirm that results have been confirmed in additional cohorts and replicates, which will help ensure reproducibility of results.

This is a broad ranging study that has been further extended in addressing the comments of the reviewers. In all, the revised manuscript contains data from 234 animals (nearly double from the original manuscript), many of which are housed for 3+ weeks to develop the OVA model of asthma; these experiments are not trivial. We aim to minimize animal use, using repeated measure designs whenever possible, because of the time required for these experiments and our ethical and legal responsibilities. All our comparisons yielded $p < 0.01$ except for borderline effects in evaluating dose responses.

With the effect size observed, power calculations demonstrate that in all of our experiments, the “n” was sufficient to obtain a power of 0.8 or greater.

We note that our judicious approach in selecting sample size is similar to and/or exceeds other 2018 papers in Nature Communications (Hsieh et al 2018. Nat Comm 9 Article 463: doi:10.1038/s41467-018-02862-4; Moon et al. 2018 Nat Comm 9 Article 401: doi:10.1038/s41467-017-02720-9; Shekhbaei et al. 2018 Nat Comm 8 Article 370: doi:10.1038/s41467-017-02723-6).

In Fig 4 (from original manuscript) where we examined the effects of LPA and TRPV1 antagonists on breathing in conscious animals, we originally used n=7. However, in analysing the data we realized we could perform a sub analysis to determine the longevity of the effect. In this sub-analysis the sample sizes dropped to 3 and 4 in each group. As this is a particularly important experiment, and heeding the reviewer’s concern, we opted to increase the overall sample size to 12 allowing a sub analysis with groups of 6 and 6 (Please see Fig 6 in revised manuscript).

Please also note, in addressing Reviewer 2 comments we have also increased the n in experiments measuring LPA in plasma from asthmatic (7 to 13) and naïve rats (7 to 15; Fig 3e and supplementary Fig 4a in revised manuscript).

3. In Figure 2 (panels h and i), it would be helpful to report the baseline and LPA-induced resistance (e.g. in the text), so the reader can appreciate the magnitude of the LPA effect in this model.

Done. These values are now reported in the figure legend.

4. In Figure 3, please clarify when the antagonists were administered in relation to Ova challenges. It would be helpful to indicated this in the figure timeline, similar to Figure 4, for consistency.

Done. We have added an arrow to Fig. 3a and Fig 5a in the revised manuscript to illustrate when antagonists were added and the following text to the legend “Antagonists were applied 30 min before the bradykinin challenge”.

5. I recognize that the main focus of this paper is AHR and not inflammation per se. Are there any readouts of inflammation that can be shown in Figure 3?

Done. We have taken two approaches to validate the inflammatory status of our model. First we calculated the Inflammation Index for the dataset described in Fig 3b from the original manuscript (see methods in revised manuscript). Second, we performed qPCR to measure gene expression of inflammatory mediators IL-4 and eotaxin on an additional cohort of asthmatic and naïve rats following bradykinin exposure. As expected, all measurements increased in the asthmatic group. These data are presented in Fig 3c and supplementary Fig 3f in the revised manuscript.

6. Since BrP-LPA acts as an autotaxin inhibitor, as well as LPAR antagonist, it could be reducing LPA generation in some models. This needs to be addressed.

Done. To address this important point, we performed additional experiments in the *en bloc* carotid body and the anesthetized *in vivo* preparations using the LPAR antagonist Ki16425 (LPAR 1,3 and weak LPAR 2 inhibitor). Ki16425 and BrP-LPA had almost identical effects. These data are presented in new Fig 1, panels f, g and h, Fig 5 panel b and supplementary Fig 1a-d in the revised manuscript.

Reviewer #2 (Remarks to the Author):

This paper uses the ovalbumin sensitized and challenged Brown Norway rat as a model of allergic asthma. As with several other published studies, the data shows that these animals have an augmented vagal reflex response. In the present study the reflex response specifically to nebulized bradykinin is augmented in the allergically inflamed animals. Bradykinin is a strong activator of vagal C-fibers in the lungs of all laboratory animals. The increased response to bradykinin is therefore consistent with the findings of others that show vagal C-fiber responsiveness is nonselectively enhanced in this model e.g., Kuo and Lai have found that the C-fibers in the lungs are hyperresponsive to adenosine, capsaicin, 5-HT agonists in this same model (J. Appl. Physiol, 2008).

We thank the reviewer for this comment. As described in more detail below, it is important to distinguish between non-asthmatic and asthmatic airways. Both have robust responses to capsaicin, but only the asthmatic lung has a robust response to allergen and/or bradykinin (Sato, et al. 1996.J. Immunol.157: 3122–3129.; Eric, et al. 2003. Br. J. Pharmacol. 138:1589–

1597.; Ehrenfeld, P. et al. 2006. *J. Leukoc. Biol.* 80:117–124.; Broadley, et al. 2010. *J. Pharmacol. Exp. Ther.* 335:681–692.; Reynolds et al. 1999. *Am. J. Respir. Crit. Care Med.* 159:431–438.; Ellis et al. 2004. *Naunyn-Schmiedeberg's Arch. Pharmacol.* 369:166–178). In the original manuscript we showed that the carotid bodies are essential for bradykinin-induced asthmatic hyper-responsiveness; we now provide new data demonstrating that the carotid bodies are not essential for the capsaicin-induced effects. Capsaicin and bradykinin therefore likely work through distinct mechanisms. Capsaicin has rapid and specific effects by stimulating a vagal-vagal reflex, whereas bradykinin has broad effects, serving as a chemoattractant (Sato, et al. 1996. *J. Immunol.* 157: 3122–3129.; Eric, et al. 2003. *Br. J. Pharmacol.* 138:1589–1597.; Ehrenfeld, P. et al. 2006. *J. Leukoc. Biol.* 80:117–124.; Broadley, et al. 2010. *J. Pharmacol. Exp. Ther.* 335:681–692.) in an inflamed lung and triggering the release of LPA into the blood stream.

General points

1. According to the authors a motivating factor of this study is to explain why carotid body resection is effective in human asthma. This seems somewhat disingenuous given the intense skepticism about any beneficial effect of carotid body resection on asthma. In fact, a position paper was published about this in *JACI* in August 1986. The authors point out that a review of the literature shows that when the patients were properly diagnosed and followed-up over time, there was no benefit of carotid body resection on asthma. Most of the studies are on unilateral resection, but the same skepticism is found in modern literature with the more dangerous bilateral resections. Therefore, I think it is misleading to presume the present data speaks to mechanisms underlying the efficacy of carotid body resection in human asthma.

Done. We agree that carotid body denervation is highly controversial as a treatment of asthma and, even if it was efficacious in treating asthma, we absolutely do not advocate for its return to clinical practice because of the risk of death via asphyxiation. The Reviewer is correct that double-blinded clinical trials have demonstrated that unilateral denervation is without clinical efficacy. There are several reports that bilateral carotid body denervation reduces asthma severity, but these reports have not been confirmed by randomized control trials. However, animal work is unequivocal for an important role of the carotid body in bronchoconstriction and asthma (Denjean et al. 1991. *Respir Physiol* 83:201–10.; Finley & Katz. 1992. *Brain Res.* 572:108–116.; Nadel & Widdicombe. 1962. *J. Physiol.* 163:13–33. Shen et al. 2012. *Am. J. Physiol.*-

Regul. Integr. Comp. Physiol. 303:R1175–R1185. Vidruk. 1985. J Appl Physiol. 59:941–6. Widdicombe. 1966. J. Physiol. 186:56–88. Iscoe & Fisher. 1995. J. Appl. Physiol. 78:117–123. Haxhiu et al. 2005. J. Appl. Physiol. 98:1961–1982. Mazzone & Canning. 2002. 2:220–228. Ahmed & Marchette. 1985. Am Rev Respir Dis 132:839–44. Xu et al. 2005. J Appl Physiol. 99:1782–8.). We have revised the manuscript to ensure this is correctly reflected.

If the paper would stick closer to the interesting basic science presented, and not so much on trying to prove asthma mechanisms, it would be an easier paper to follow. Bradykinin nebulization is referred to as “asthmatic provocation” in several places. The second sentence of the discussion referred to blocking LPA signaling as ameliorating acute asthmatic bronchoconstriction...” when I think “asthmatic bronchoconstriction” here refers to nebulized bradykinin-induced cholinergic reflex activity. The title refers to acute asthma attacks, when no such thing is studied, etc..

Done. We are intent on reaching a balance between scientific precision and the needs of the broad readership of Nature Communications. We have modified the body of the text to try and meet both goals and will consult the Editor regarding the wording of the title.

3. A novel and very interesting aspect of this study is that it argues that the enhanced parasympathetic reflex response to nebulized bradykinin in allergically inflamed rat airways is abolished by acute denervation of the carotid bodies.

Correct.

This would mean that activity in the carotid sinus nerve is somehow required for the classic and well studied airway vagal C-fiber induced vagal parasympathetic bronchoconstrictor reflex.

Incorrect. The reviewer is assuming that bradykinin is working exclusively through a C-fibre vagal-vagal reflex. In fact, bradykinin serves as a chemoattractant and stimulates mast cells that are in particular abundance in the asthmatic lung (Sato, et al. 1996. J. Immunol. 157: 3122–3129.; Eric, et al. 2003. Br. J. Pharmacol. 138:1589–1597.; Ehrenfeld, P. et al. 2006. J. Leukoc. Biol. 80:117–124.; Broadley, et al. 2010. J. Pharmacol. Exp. Ther. 335:681–692.).

To demonstrate that the C-fibre induced vagal parasympathetic bronchoconstrictor reflex is independent of the carotid body we performed a new set of experiments.

We challenged naïve carotid body denervated animals with capsaicin whilst measuring airway resistance using the Flexivent system. In one group of animals the vagus was intact and in another it was denervated. Capsaicin caused increased airway resistance in vagal intact but not vagotomised rats. Thus, the C-fibre mediated vagal-vagal reflex **does not** require the carotid bodies. This is in sharp contrast to the effects of nebulised bradykinin which is critically dependent on an intact carotid body (see Fig 5 d and supplementary Fig 5a, b in revised manuscript).

This would be very important, but also very peculiar. How do the authors imagine such a phenomenon occurring?

Capsaicin and bradykinin likely work through distinct mechanisms. Capsaicin has rapid and specific effects by stimulating a vagal-vagal reflex, whereas bradykinin has broad effects, serving as a chemoattractant in an inflamed lung (Sato, et al. 1996. *J. Immunol.* 157: 3122–3129.; Eric, et al. 2003. *Br. J. Pharmacol.* 138:1589–1597.; Ehrenfeld, P. et al. 2006. *J. Leukoc. Biol.* 80:117–124.; Broadley, et al. 2010. *J. Pharmacol. Exp. Ther.* 335:681–692.). Our data show that these effects include triggering the release of LPA into the blood stream. This then activates the carotid body causing parasympathetic activation.

Please note most studies in asthma have utilized anesthetized preparations to assess airway responsiveness. Choice of anesthetic may be an important consideration. For example, one study (Bonora & Vizek 1999. *J Appl Physiol.* 87:15-21) utilizing sodium pentobarbital did not find a role of carotid bodies in bronchoconstriction, whereas multiple studies using conscious animals, urethane or decerebrate preparations did (Nadel & Widdicombe 1962. *J Physiol* 163:13-33.; Mitchell et al. 1985. *J Appl Physiol* 58:911-920.; Iscoe and Fischer 1995 *J Appl Physiol.* 78:117-123; Fischer et al 1987. *Can J Physiol Pharmacol.* 1987. 65:1234-1238; Denjean et al. 1991 *Resp Physiol.* 83:201-10.; Vidruk and Sorkness 1985. *Am Rev Resp Dis.* 132:287-291). We used alfaxan which is known to maintain all autonomic reflexes (Marshall & Metcalfe 1990. *J Physiol* 426:335-353 and references therein).

With regards to the carotid body driving parasympathetic activity in general and bronchoconstriction specifically, this is strongly supported in the literature

(Denjean et al. 1991. *Respir Physiol* 83:201–10.; Finley & Katz. 1992. *Brain Res.* 572:108–116.; Nadel & Widdicombe. 1962. *J. Physiol.* 163:13–33. Shen et al. 2012. *Am. J. Physiol.-Regul. Integr. Comp. Physiol.* 303:R1175–R1185. Vidruk. 1985. *J Appl Physiol.* 59:941–6. Widdicombe. 1966. *J. Physiol.* 186:56–88. Iscoe & Fisher. 1995. *J. Appl. Physiol.* 78:117–123. Haxhiu et al. 2005. *J. Appl. Physiol.* 98:1961–1982. Mazzone & Canning. 2002. 2:220–228. Ahmed & Marchette. 1985. *Am Rev Respir Dis* 132:839–44. Xu et al. 2005. *J Appl Physiol.* 99:1782–8.).

The data presented do not make for a compelling case. To draw such a bold conclusion, more work is required. Is it possible that there is some artefactual effect of acutely ablating the carotid sinus nerves such that all vagal reflexes are blocked? Acutely severing inputs to the brainstem can have transient and profound effects. For example, bilateral severing of the vagus nerves in rodents can lead acutely to a complete loss of respiratory control and death. The hypothesis at hand is that the carotid bodies are needed for bradykinin-induced reflexes in allergically inflamed lungs. Might this be better addressed with a model whereby the carotid sinus nerves are cut in a survival surgery procedure. Then after the animals have had sufficient time to recover (~a week) one can determine if augmented vagal cholinergic airway reflexes are still blocked?

Done. To address this comment, we performed additional experiments that involved asthmatic rats with chronically denervated carotid bodies. 4-5 days after denervation, we measured airway resistance in anesthetized rats using the Flexivent system. As in the acutely denervated preparation, bradykinin caused a pronounced increase in airway resistance in sham asthmatic rats but not in chronically denervated asthmatic rats (please see Fig 5c in the revised manuscript).

4. In the present study bradykinin, surprisingly, only evoked a reflex bronchoconstriction in allergically inflamed lungs?

Correct. Please see Fig 3d, 5b in revised manuscript. That bradykinin produced very little response in our naïve animals is consistent with animal (Ellis et al. 2004. *Naunyn. Schmiedebergs Arch. Pharmacol.* 369:166–178. Hannon et al. 2001. *Br. J. Pharmacol.* 132:1509–1523.) and human data (Reynoldset al. 1999. *Am. J. Respir. Crit. Care Med.* 159:431–438.) showing that C-fibre activation by bradykinin is minimal in non-asthmatic animals and humans.

Does acute carotid sinus nerve ablation also prevent vagal airway reflexes in naïve animals. In other words, does acute severing of the carotid bodies abolish airway reflexes evoked by a stimulus that does not require airway inflammation to manifest itself?

As stated above, we challenged naïve carotid body denervated animals to capsaicin whilst measuring airway resistance using the Flexivent system. In one group of animals the vagus was intact and in another it was denervated. Capsaicin caused increase airway resistance in vagal intact but not vagotomised rats. Thus, the C-fibre mediated vagal-vagal reflex **does not** require the carotid bodies.

These data fit with Fig 2 h and i, in that LPA infusion in naïve animals produced carotid body-mediated bronchoconstriction and was lost with denervation.

5. The hypothesis here is that OVA leads to (a very modest) increase in plasma LPA and this is a necessary precondition for bradykinin applied to the lungs to evoke a bronchoconstrictor reflex (a reflex that somehow is dependent on activity in the carotid bodies).

With regards to the physiological relevance of the increase in plasma LPA, as reported above, we developed a new bio-assay to test if plasma containing asthma-elevated LPA was sufficient to increase carotid body activity. We took plasma from asthmatic rats 3hr after OVA challenge and perfused it through the *en bloc* carotid body preparation (1 ml of plasma in 100ml of perfusate; see revised manuscript for details). Plasma from asthmatic rats increased carotid sinus nerve (CSN) activity by ~45% (plasma from naïve rats increased CSN activity by less than ~15%; Fig 4a, b in revised manuscript). Moreover, LPA_R and TRPV1 antagonists abolished this increase. This experiment demonstrates the increase in plasma LPA is sufficient to increase activity of the carotid body.

With regards to carotid body activity driving parasympathetic activity in general and bronchoconstriction specifically, this is strongly supported in the literature (Denjean et al. 1991. *Respir Physiol* 83:201–10.; Finley & Katz. 1992. *Brain Res.* 572:108–116.; Nadel & Widdicombe. 1962. *J. Physiol.* 163:13–33. Shen et al. 2012. *Am. J. Physiol.-Regul. Integr. Comp. Physiol.* 303:R1175–R1185. Vidruk. 1985. *J Appl Physiol.* 59:941–6. Widdicombe. 1966. *J. Physiol.* 186:56–88. Iscoe & Fisher. 1995. *J. Appl. Physiol.* 78:117–123. Haxhiu et al. 2005. *J. Appl. Physiol.* 98:1961–1982. Mazzone & Canning.

2002. 2:220–228. Ahmed & Marchette. 1985. Am Rev Respir Dis 132:839–44. Xu et al. 2005. J Appl Physiol. 99:1782–8.). We confirmed this pathway using the dual perfused *in situ* and anesthetized preparations; carotid body activity causes an increase in vagal activity and increased airway resistance (Fig 2).

This conclusion is based only on the observation that BrP-LPA a potent LPA-receptor antagonist (with other actions such as phospholipase and phosphodiesterase inhibitory activity) abolished the ability of bradykinin to evoke a cholinergic reflex. Does BRP-LPA cross the BBB where it may interfere with neurotransmission? Perhaps phospholipase is needed for bradykinin to activate C-fiber terminals?

Done. As reported above, to address this important point, we performed additional experiments in the *en bloc* carotid body and the anesthetized *in vivo* preparations using the LPAR antagonist Ki16425 (LPA 1,3 and weak2 inhibitor). Ki16425 is not a phosphodiesterase inhibitor. Ki16425 and BrP-LPA had almost identical effects. These data are now added in Fig 1, panels f, g and h, Fig 5 panel b and supplementary Fig 1a-d in the revised manuscript.

Although the MS states bradykinin increases serum LPA in the OVA rats, the data in figure 3vi do not support this assertion. The baseline serum LPA is slightly elevated by OVA, but bradykinin does not seem to further enhance the LPA concentration.

Done. As this was an important experiment and to comply with the spirit of Reviewer 1's request to consider group sizes we increased the number of animals in these experiments from a total of 14 to 28 animals. Analysis of the new data demonstrates a significant increase in LPA in response to bradykinin in OVA-treated animals (3.7 ± 1.1 ; $P < 0.001$). Figure 3e is updated accordingly along with a new supplementary Figure 4.

Is the LPA increased by the repetitive allergen challenge is somehow required for bradykinin to activate the airway C-fibers? Or is the argument that bradykinin effectively activated the airway C-fibers but can only lead to a cholinergic reflex if LPA in the serum facilitates carotid body activity. This is very confusing to me—it would help if the authors stated their specific hypothesis and then presented the data in its favor.

Done. In the revised manuscript we have refined hypotheses and discussed both the carotid body-vagal reflex and the C-fibre vagal-vagal reflex with regards to their independence.

At a minimum one would wonder if more selective LPA₁ antagonists that are not enzyme inhibitors mimic the effect of BrP-LPA.

Done. Ki16425 had similar effects to Brp-LPA. Please see above for details.

Since LPA receptors are expressed by the petrosal neurons, it would seem likely that they are also expressed nodose neurons. The vagal C-fibers terminate in the epithelium therefore these nerve endings are likely to come in contact with the largest concentration of the LPA—presuming, as stated, that OVA is stimulating LPA release from airway epithelium.

Without disagreeing with these observations, we note that carotid body denervation abolished the ability of LPA to increase vagal activity in the *in situ* dual perfused preparation (Fig 2a-f). We also note that the effects of bradykinin on airway resistance in asthmatic (Fig 3d, 5b revised manuscript) and naïve anesthetized rats in response to intravenous LPA infusion (Fig 2h, i) is abolished by carotid body denervation. Therefore, parsimony would suggest that the carotid bodies are the main site of action.

Does nebulized LPA into the lungs evoke the reflex, as you argue that it will when infused to the CBs? If so is this airway reflex also blocked by lesioning the carotid sinus nerves. Can infused LPA into the lungs enhance the bradykinin evoked reflex or does it indeed need to reach the CBs?

Done. To address this comment, we performed additional experiments in which we nebulized LPA into the lung of anesthetized asthmatic rats whilst measuring airway resistance using the Flexivent. Increase in airway resistance in response to LPA inhalation occurred after 30 min, congruent with a previous investigation (Hashimoto et al. 2001. Life Sci 70, 199–205). Again, carotid body denervation minimized the increase in airway resistance. These data are consistent with a downstream effect of LPA on the carotid body. Please see the revised manuscript (Supplementary Fig 5c in the revised manuscript).

Reviewers' comments:

Reviewer #1 (Remarks to the Author):

The authors have been quite responsive in the revised manuscript and rebuttal. New data are included: (1) demonstrating that plasma from asthmatic rats increased CSN activity in an LPAR- and TRPV1-dependent manner (Fig. 4), (2) validating the ELISA (Suppl. Fig. 4), and (3) demonstrating an effect of Ki16425 (in addition to Brp_LPA). These results strengthen their conclusions, and support the overall working model. I have a few comments and remaining concerns.

1. It is disappointing that the authors did not attempt to measure LPA concentrations or species using LC/MS. Notwithstanding the concerns about potential artifacts introduced during extraction in the rebuttal, LC/MS remains the gold standard for lysolipid analyses and quantitation.

2. It would be helpful in Figure 6 cartoon to indicate when the antagonists were administered (similar to other figures).

3. Lines 271-278: The statements that "During allergen challenge, several species of LPA are released by lung epithelial cells into surrounding tissue...", and "...LPA is also released systemically, in arterial plasma" are not entirely correct and poorly worded. LPA is present in both circulation and also in lung fluids as sampled by BAL. The LPA species in these compartments are different, and although lung LPA levels increase after allergen challenge, it is not so clear that plasma levels also increase. In reference 30 (Park et al), the increase in serum LPA after allergen challenge was trivial. Whether LPA leaks between these different compartment (plasma and lung), or is generated de novo in different tissues, is not clear and an area of active research. This section should be re-worded and the references updated. Specific revisions should include and/or address the following.

- Reference 60 appears to be to a conference abstract and should be removed
- Two human studies have shown that after allergen challenge, local concentrations of LPA increase in BAL fluids (Park et al, reference 30, and PMID 17359381 should also be included here)
- In Park et al (reference 30) the increase in plasma LPA after allergen challenge was very small. The authors should acknowledge this, and point out that more research about changes in plasma LPA levels in human subjects after environmental exposures is needed in order to fully put their results in proper context.
- Extracellular LPA can derive from phosphatidic acid (via the action of phospholipase) or from catalysis of LPC by the enzyme autotaxin (e.g. PMID 12354767)
- LPA circulates bound to albumin and also autotaxin, and has a very short half-life due to the action of lipid phosphate phosphatases (LPP's) including LPP1 (PMID 19215222). Therefore, it is difficult to extrapolate from measurements of LPA to its true bioactivity in vivo.
- Factors responsible for increased LPA levels during inflammation (or in response to bradykinin in their model) are not known, and could reflect enhanced de novo generation or reduced breakdown.

Reviewer #2 (Remarks to the Author):

No further comments

Reviewers' comments:

Reviewer #1 (Remarks to the Author):

The authors have been quite responsive in the revised manuscript and rebuttal. New data are included: (1) demonstrating that plasma from asthmatic rats increased CSN activity in an LPAR- and TRPV1-dependent manner (Fig. 4), (2) validating the ELISA (Suppl. Fig. 4), and (3) demonstrating an effect of Ki16425 (in addition to Brp_LPA). These results strengthen their conclusions, and support the overall working model. I have a few comments and remaining concerns.

1. It is disappointing that the authors did not attempt to measure LPA concentrations or species using LC/MS. Notwithstanding the concerns about potential artifacts introduced during extraction in the rebuttal, LC/MS remains the gold standard for lysolipid analyses and quantitation.

As suggested by Reviewer 1 and requested by the Editor we used LC/MS to measure LPA concentration in asthma plasma. We used our remaining frozen plasma samples extracted at the end of the experimental protocol in Fig 3 (the only point at which we could fully exsanguinate the animal). These new data confirm LPA in asthma plasma, ~20 min after challenge, is in the low micromolar range ($1.5 \pm 0.2 \mu\text{M}$).

The concentration measured with LC/MS is slightly lower than that measured with ELISA ($8.4 \pm 1.7 \mu\text{M}$). However, the relationship between LC/MS and ELISA measurements of the same samples was $R = 0.8$ ($P = 0.005$). Confirming the quantitative capability of ELISA.

Our previous data using the *en bloc* carotid body preparation demonstrated $2.5 \mu\text{M}$ of LPA significantly increases carotid body activity; to ensure the physiological effect of LPA extends down to $1.5 \mu\text{M}$ we also add new data demonstrating $1.5 \mu\text{M}$ increases carotid body activity (supplementary Fig 5).

Please note that we used LC/MS as suggested by the reviewer notwithstanding the following important points:

- (a) That ELISA is used broadly in research and in FDA approved assays. Looking through Nature Communications articles over the last year we counted 88 that used ELISA, 4 of which measured lipids; none of these validated the ELISA with LC/MS (please see list below; lipids starred). LC/MS **and** ELISA are accepted methods of measuring LPA as illustrated in this review: JESIONOWSKA et al 2014 (PMID: 24613261).
- (b) Both ELISA and LC/MS are powerful techniques but both have limitations. For example, some have raised concerns about potential artifacts introduced during extraction of LPA for LC/MS analysis as described in the initial rebuttal and acknowledged by Reviewer 1 (PMID: 24921707). Another limitation of LC/MS is the dependence on known standards. There are 5 species of LPA commercially available as standards, yet 22 species of LPA were identified. Therefore, the

calculation of the concentration of each individual species without standard and, consequently total LPA may be over or underestimated.

- (c) In our initial responses to Reviewer 1, we provided a literature search which demonstrates that several forms of MS have been used, and many of them yielded LPA concentrations in the μM range (<https://www.dropbox.com/s/m92rt4vwwj89p29/LPA%20concentration%20meta%20analyses.xlsx?dl=0>), the same range we detected with ELISA. Essentially, our new LC/MS data are consistent with these measurements.
- (d) In our revisions, we validated the ELISA kit we used by spiking plasma from non-asthmatic rats with known concentrations of LPA (supplementary Fig 4b). These experiments were acknowledged but not challenged by Reviewer 1.
- (e) In our revisions, we conducted a bioassay with our *en bloc* preparation to demonstrate that the plasma from challenged asthmatic rats increases carotid sinus nerve activity much more than plasma from non-asthmatic rats, and this difference was LPA receptor/TRPV1 dependent (Figure 4). Thus, regardless of the difficulty in measuring the precise concentration of LPA in plasma because of its short half-life (~5 min; PMID: 23948545) and multiple mechanisms of degradation as described by Reviewer 1, the concentration of LPA in asthma plasma has biological effects on the carotid body.
- (f) Both our ELISA and LC/MS data may underestimate the peak LPA concentration following asthmatic challenge because blood was extracted ~20 min after challenge. We have yet to determine the dynamics of LPA release and accumulation following challenge.

2. It would be helpful in Figure 6 cartoon to indicate when the antagonists were administered (similar to other figures).

Done.

3. Lines 271-278: The statements that "During allergen challenge, several species of LPA are released by lung epithelial cells into surrounding tissue...", and "...LPA is also released systemically, in arterial plasma" are not entirely correct and poorly worded. LPA is present in both circulation and also in lung fluids as sampled by BAL. The LPA species in these compartments are different, and although lung LPA levels increase after allergen challenge, it is not so clear that plasma levels also increase. In reference 30 (Park et al), the increase in serum LPA after allergen challenge was trivial. Whether LPA leaks between these different compartment (plasma and lung), or is generated *de novo* in different tissues, is not clear and an area of active research. This section should be re-worded and the references updated. Specific revisions should include and/or address the following.

We thank the reviewer for these suggestions. Given our data we consider it highly likely that LPA stimulation of the carotid body originates from LPA released from the lung. However, acknowledging this reviewer's concerns and the fact that we have not explicitly determined that the lung is the source of increased LPA, we have

subtly revised the manuscript accordingly. We have changed the wording of the title, modified the abstract and implemented the suggested changes below.

- Reference 60 appears to be to a conference abstract and should be removed

Done.

- Two human studies have shown that after allergen challenge, local concentrations of LPA increase in BAL fluids (Park et al, reference 30, and PMID 17359381 should also be included here)

Done.

- In Park et al (reference 30) the increase in plasma LPA after allergen challenge was very small. The authors should acknowledge this, and point out that more research about changes in plasma LPA levels in human subjects after environmental exposures is needed in order to fully put their results in proper context.

Done.

- Extracellular LPA can derive from phosphatidic acid (via the action of phospholipase) or from catalysis of LPC by the enzyme autotaxin (e.g. PMID 12354767)

Done.

- LPA circulates bound to albumin and also autotaxin, and has a very short half-life due to the action of lipid phosphate phosphatases (LPP's) including LPP1 (PMID 19215222). Therefore, it is difficult to extrapolate from measurements of LPA to it's true bioactivity in vivo.

Done.

- Factors responsible for increased LPA levels during inflammation (or in response to bradykinin in their model) are not known, and could reflect enhanced de novo generation or reduced breakdown.

Done.

Reviewer #2 (Remarks to the Author):

No further comments

Search: "ELISA", Journal: Nature Communications, 2018-2018

- *****1. Nat Comm 9. Article number 279 (2018) doi:10.1038/s41467-017-02648-0
ELISA for: testosterone
- *****2. Nat Comm 9. Article number 890 (2018) doi:10.1038/s41467-018-03196-x
ELISA for: adiponectin
- *****3. Nat Comm 9. Article number 272 (2018) doi:10.1038/s41467-017-02677-9
ELISA for: FG21 and adiponectin
- *****4. Nat Comm 9. Article number 1674 (2018) doi:10.1038/s41467-018-04048-4
ELISA for: Pcsk9
5. Nat Comm 9. Article number 1037 (2018) doi:10.1038/s41467-018-03439-x
ELISA for: hyaluronan
6. Nat Comm 9. Article number 291 (2018) doi:10.1038/s41467-017-02533-w
ELISA for: amyloid β
7. Nat Comm 9. Article number 525 (2018) doi:10.1038/s41467-018-02896-8
ELISA for: amyloid, c reactive protein
8. Nat Comm 9. Article number 55 (2018) doi:10.1038/s41467-017-02490-4
ELISA for: collagen (CTX-I)
9. Nat Comm 9, Article number: 1784 (2018) doi:10.1038/s41467-018-04120-z
ELISA for: IL-6, amyloid β , C/EBP β and C/EBP α
10. Nat Comm 9, Article number: 1800 (2018): doi:10.1038/s41467-018-04238-0
ELISA for: amyloid β
11. Nat Comm 9, Article number: 1802 (2018): doi:10.1038/s41467-018-04255-z
ELISA for: amyloid β
12. Nat Comm 9. Article number: 1646(2018) doi:10.1038/s41467-018-03773-0
ELISA for: Leptin, Insulin, CCL-20/MIP-3, albumin
13. Nat Comm 9. Article number: 1479 (2018) doi:10.1038/s41467-018-03674-2
ELISA for: amyloid β
14. Nat Comm 9, Article number: 1560 (2018) doi:10.1038/s41467-018-03669-z
ELISA for: IL-1 β , IL-18, TNF, IL-6, RANTES, and CXCL1
15. Nat Comm 9, Article number: 1230 (2018) doi:10.1038/s41467-018-03662-6
ELISA for: anti-ZikV and anti CHIKV
16. Nat Comm 9, Article number: 1743 (2018) doi:10.1038/s41467-018-04172-1
ELISA for: prostrate-specific antigen, carcinoembryonic antigen and α -fetoprotein
17. Nat Comm 9, Article number: 1421 (2018) doi:10.1038/s41467-018-03636-8
ELISA for: IgE and IgG
18. Nat Comm 9, Article number: 1628 (2018) doi:10.1038/s41467-018-04063-5
ELISA for: IgM, IgG1, IgG2c, IgG2b, IgG3, IFN- γ , IL-21, IL-4, IL-6, and IL-2
19. Nat Comm 9, Article number: 1734 (2018) doi:10.1038/s41467-018-04092-0
ELISA for: IL-8, CXCL-1
20. Nat Comm 9, Article number: 1603 (2018) doi:10.1038/s41467-018-03886-6
ELISA for: IL-4, 5, 6, 13, 33
21. Nat Comm 9, Article number: 1136 (2018) doi:10.1038/s41467-018-03530-3
ELISA for: IgG (specific for 3 vaccines)
22. Nat Comm 9, Article number: 1685 (2018) doi:10.1038/s41467-018-03966-7
ELISA for: CXCL1, CXCL2 CXCL5

23. Nat Comm 9. Article number: 1855 (2018) doi:10.1038/s41467-018-04175-y
ELISA for: vegf-a.
24. Nat Comm 9. Article number: 1759(2018) doi:10.1038/s41467-018-03907-4
ELISA for: Calmodulin
25. Nat Comm 9. Article number: 1306 (2018) doi:10.1038/s41467-018-03755-2
ELISA for: Insulin receptor
26. Nat Comm 9. Article Number: 863 (2018) doi:10.1038/s41467-018-03318-5
ELISA for: IgA, IL-1 β , IL-6, IL-13, IL-23, IFN- γ , TNF, IL-17A, IL-22
27. Nat Comm 9. Article number: 106 (2018) doi:10.1038/s41467-017-02645-3
ELISA for: IL-1 β
28. Nat Comm 9. Article number: 741 (2018) doi:10.1038/s41467-017-02696-6
ELISA for: CTLA-4, TGF β , PD-L1 antibodies
29. Nat Comm 9. Article number: 996 (2018) doi:10.1038/s41467-018-03409-3
ELISA for: IL-1 α , IL-1 β , IL-6, and TNF
30. Nat Comm 9. Article number: 1485 (2018) doi:10.1038/s41467-018-03782-z
ELISA for: IL-29/IL-28B, IFN- α , IFN- β , TNF- α
31. Nat Comm 9. Article number 1199 (2018) doi:10.1038/s41467-018-03323-8
ELISA for: BAFFR-Fc, BCMA-Fc, or TACI-Fc
32. Nat Comm 9. Article number 1431 (2018) doi:10.1038/s41467-018-03627-9
ELISA for: Heat shock protein
33. Nat Comm 9. Article number 621 (2018) doi:10.1038/s41467-018-03061-x
ELISA for: FcRN, IgG
34. Nat Comm 9. Article number 242 (2018) doi:10.1038/s41467-017-02682-y
ELISA for: IL-1 α , IL-1 β , IL-18, TNF
35. Nat Comm 9. Article number 1752 (2018) doi:10.1038/s41467-018-04038-6
ELISA for: Dickkopf 3
36. Nat Comm 9. Article number 1758 (2018) doi:10.1038/s41467-018-03750-7
ELISA for: IgG
37. Nat Comm 9. Article number 613 (2018) doi:10.1038/s41467-018-02936-3
ELISA for: IFN α & β
38. Nat Comm 9. Article number 558 (2018) doi:10.1038/s41467-017-02646-2
ELISA for: Pfs48/45, Pfs230, 43
39. Nat Comm 9. Article number 1420 (2018) doi:10.1038/s41467-018-03704-z
ELISA for: IL-12, IL-1 β , mKC, and IL-17F
40. Nat Comm 9. Article number 1379 (2018) doi:10.1038/s41467-018-03847-z
ELISA for: specific antibodies.
41. Nat Comm 9. Article number 1243 (2018) doi:10.1038/s41467-018-03563-8
ELISA for: IFN- β , TNF- α , IL-6
42. Nat Comm 9. Article number 1523 (2018) doi:10.1038/s41467-018-03925-2
ELISA for: CD40
43. Nat Comm 9. Article number 1006 (2018) doi:10.1038/s41467-018-03455-x
ELISA for: IgA
44. Nat Comm 9. Article number 672 (2018) doi:10.1038/s41467-018-03051-z
ELISA for: NF- κ B, IL-1 β , CXCL1, RelA, RelB, c-Rel, P50, and P52 DNA-binding activity
45. Nat Comm 9. Article number 1591 (2018) doi:10.1038/s41467-018-03900-x

- ELISA for: HMGB1
46. Nat Comm 9. Article number 551 (2018) doi:10.1038/s41467-018-02988-5
ELISA for: TGF- β 1, PDGF-BB
 47. Nat Comm 9. Article number 503 (2018) doi:10.1038/s41467-017-02731-6
ELISA for: Chi311 protein
 48. Nat Comm 9. Article number 1102 (2018) doi:10.1038/s41467-018-03495-3
ELISA for: multiple interleukins
 49. Nat Comm 9. Article number 816 (2018) doi:10.1038/s41467-018-03105-2
ELISA for: troponin
 50. Nat Comm 9. Article number 463 (2018) doi:10.1038/s41467-018-02862-4
ELISA for: multiple interleukins
 51. Nat Comm 9. Article number 1371 (2018) doi:10.1038/s41467-018-03762-3
ELISA for: p27 antigen
 52. Nat Comm 9. Article number 725 (2018) doi:10.1038/s41467-018-03129-8
ELISA for: interleukins, chemoattractants and IgG
 53. Nat Comm 9. Article number 1021 (2018) doi:10.1038/s41467-018-03470-y
ELISA for: TNF, TGF, interleukins
 54. Nat Comm 9. Article number 1296 (2018) doi:10.1038/s41467-018-03692-0
ELISA for: VEGF
 55. Nat Comm 9. Article number 1325 (2018) doi:10.1038/s41467-018-03787-8
ELISA for: EPO
 56. Nat Comm 9. Article number 1241 (2018) doi:10.1038/s41467-018-03584-3
ELISA for: hepatic cellular carcinoma markers
 57. Nat Comm 9. Article number 792 (2018) doi:10.1038/s41467-018-03226-8
ELISA for: Integrins
 58. Nat Comm 9. Article number 1837 (2018) doi:10.1038/s41467-018-04221-9
ELISA for: hepatocyte factor IX
 59. Nat Comm 9. Article number 1165 (2018) doi:10.1038/s41467-018-03544-x
ELISA for: HIV p21
 60. Nat Comm 9. Article number 1196 (2018) doi:10.1038/s41467-018-03625-x
ELISA for: T3SS chaperone CeST and CsrA
 61. Nat Comm 9. Article number 873 (2018) doi:10.1038/s41467-018-03225-9
ELISA for: Interferons
 62. Nat Comm 9. Article number 575 (2018) doi:10.1038/s41467-018-03079-1
ELISA for: Prolyl hydroxylases
 63. Nat Comm 9. Article number 1416 (2018) doi:10.1038/s41467-018-03672-4
ELISA for: GDF2
 64. Nat Comm 9. Article number 7 (2018) doi:10.1038/s41467-017-02312-7
ELISA for: IgG, CD23
 65. Nat Comm 9. Article number 1350 (2018) doi:10.1038/s41467-018-03853-1
ELISA for: Insulin
 66. Nat Comm 9. Article number 355 (2018) doi:10.1038/s41467-017-02610-0
ELISA for: TNF, IFN γ and VEGF-A
 67. Nat Comm 9. Article number 341 (2018) doi:10.1038/s41467-017-02661-3
ELISA for: β -arrestin
 68. Nat Comm 9. Article number 636 (2018) doi:10.1038/s41467-018-03038-w

- ELISA for: FGF21
69. Nat Comm 9. Article number 682 (2018) doi:10.1038/s41467-018-02969-8
ELISA for: HBsAg and HBeAg, albumin
 70. Nat Comm 9. Article number 1074 (2018) doi:10.1038/s41467-018-03473-9
ELISA for: heat shock protein
 71. Nat Comm 9. Article number 359 (2018) doi:10.1038/s41467-017-02725-4
ELISA for: custom/specific antibodies
 72. Nat Comm 9. Article number 1645 (2018) doi:10.1038/s41467-017-01240-w
ELISA for: Incretins
 73. Nat Comm 9. Article number 263 (2018) doi:10.1038/s41467-017-02499-9
ELISA for: ZIKV antibodies
 74. Nat Comm 9. Article number 1281 (2018) doi:10.1038/s41467-018-03668-0
ELISA for: cGMP
 75. Nat Comm 9. Article number 1251 (2018) doi:10.1038/s41467-018-03632-y
ELISA for: HIV glycan antibodies
 76. Nat Comm 9. Article number 1513 (2018) doi:10.1038/s41467-018-03986-3
ELISA for: IFN- γ , TNF- α , IL-10, and IL-4
 77. Nat Comm 9. Article number 258 (2018) doi:10.1038/s41467-017-02747-y
ELISA for: anti glycan antibodies
 78. Nat Comm 9. Article number 53 (2018) doi:10.1038/s41467-017-02242-4
ELISA for: histones
 79. Nat Comm 9. Article number 113 (2018) doi:10.1038/s41467-017-02488-y
ELISA for: GLP 1
 80. Nat Comm 9. Article number 367 (2018) doi:10.1038/s41467-017-02664-0
ELISA for: Insulin
 81. Nat Comm 9. Article number 177 (2018) doi:10.1038/s41467-017-02539-4
ELISA for: Insulin
 82. Nat Comm 9. Article number 528 (2018) doi:10.1038/s41467-018-02827-7
ELISA for: Fab 1A12
 83. Nat Comm 9. Article number 507 (2018) doi:10.1038/s41467-017-02578-x
ELISA for: cAMP
 84. Nat Comm 9. Article number 132 (2018) doi:10.1038/s41467-017-02542-9
ELISA for: HIV 9 24
 85. Nat Comm 9. Article number 85 (2018) doi:10.1038/s41467-017-02611-z
ELISA for: interferons
 86. Nat Comm 9. Article number 30 (2018) doi:10.1038/s41467-017-02537-6
ELISA for: insulin
 87. Nat Comm 9. Article number 11 (2018) doi:10.1038/s41467-017-02401-7
ELISA for: MCP-1 IL6
 88. Nat Comm 9. Article number 12 (2018) doi:10.1038/s41467-017-02416-0
ELISA for: α -synuclein

REVIEWERS' COMMENTS:

Reviewer #1 (Remarks to the Author):

The authors made a good faith effort to respond to the critique of the revised manuscript. Additional analyses of LPA by MS revealed lower concentrations than by ELISA, but their Supporting Figure 5 suggest that lower concentrations than originally studied have the same effect on carotid body reflexes. The revisions to the discussion clarify how and where LPA might be generated and act in humans with asthma, and enhance the translational impact of the paper.

REVIEWERS' COMMENTS:

Reviewer #1 (Remarks to the Author):

The authors made a good faith effort to respond to the critique of the revised manuscript. Additional analyses of LPA by MS revealed lower concentrations than by ELISA, but their Supporting Figure 5 suggest that lower concentrations than originally studied have the same effect on carotid body reflexes. The revisions to the discussion clarify how and where LPA might be generated and act in humans with asthma, and enhance the translational impact of the paper.

We thank the reviewer for their appreciation and their hard work throughout the review process.